# The nucleus accumbens shell regulates hedonic feeding via a rostral hotspot

Alina-Măriuca Marinescu[1]*, Eshita Kamal[1], Peter Leary[2], Keila Navarro I Batista[3], Manuel Klug[1], Nataša Savić[4], Christelle Le Foll[3], Marie A Labouesse[1,5]*

[1]Department of Health Sciences and Technology, ETH Zurich, Zurich, Switzerland; [2]Functional Genomics Center Zurich, University of Zurich/ETH Zurich, Zurich, Switzerland; [3]Institute of Veterinary Physiology, University of Zurich – Vetsuisse, Zurich, Switzerland; [4]ETH Phenomics Center, ETH Zurich, Zurich, Switzerland; [5]Neuroscience Center Zurich, University and ETH Zurich, Zurich, Switzerland

## eLife Assessment

This study provides a **valuable** contribution to understanding the functional and molecular organization of the medial nucleus accumbens shell in feeding behavior. Through a multimodal approach that integrates in vivo imaging, optogenetic manipulation, and genetic strategies, the authors present **convincing** evidence for rostro-caudal differences in D1-SPN activity, advancing and refining earlier pharmacological frameworks. The discovery of Stard5 and Peg10 as regionally informative markers, together with the introduction of a Stard5-Flp driver line, establishes a foundation for more targeted circuit dissection. While an expanded characterization of other Stard5-positive cell populations (e.g., D2-SPNs, interneurons) would strengthen the work, the experimental rigor and internal consistency of the findings are clear. Overall, this is a technically strong and conceptually meaningful study with broad relevance for those investigating neural mechanisms of reward, affect, and feeding.

*For correspondence:
alinamariuca.marinescu@hest.
ethz.ch (A-MM);
marie.labouesse@gmail.com
(MAL)

Competing interest: The authors declare that no competing interests exist.

**Abstract** The medial nucleus accumbens shell (medNAcSh) is a key regulator of hedonic feeding, controlling reward consumption through its projections to downstream structures. Recent studies showed that the primary cellular mediators of these effects are dopamine 1 receptor-positive striatal projection neurons (D1-SPNs). Specifically, D1-SPN activity gets inhibited during reward consumption, and such inhibition is necessary and sufficient to authorize consumption, independent of metabolic need. Anatomically, the medNAcSh spans 1–1.5 mm along the rostro-caudal axis in mice, and previous studies have reported functional gradients along this axis. For instance, pharmacological studies have suggested that rostral rather than caudal medNAcSh regulates appetitive behavior. However, the mechanisms underlying this topographical gradient remain unknown. Here, we hypothesized that D1-SPNs contribute to this gradient by regulating hedonic feeding via a specific hotspot in the rostral medNAcSh. Using calcium monitoring with fiber photometry in mice, we show that rostral medNAcSh D1-SPNs demonstrate inhibitory responses during reward consumption, while caudal D1-SPNs do not. Consistently, optogenetic stimulation of rostral D1-SPNs inhibits consumption, while stimulation of caudal D1-SPNs had minimal effects, confirming the existence of a functional rostro-caudal gradient. Importantly, we observed no differences between rostral and caudal D1-SPNs in their responses to aversive stimuli, suggesting that the D1-SPN gradient is specific to appetitive contexts. To investigate potential molecular correlates of this functional gradient, we leveraged open-source anatomy datasets and performed fluorescent in situ hybridization, identifying *Stard5* and *Peg10* as markers enriched in the rostral and caudal medNAcSh, respectively. Finally, we developed a novel Stard5-Flp driver line to selectively target the rostral hotspot and demonstrated that Stard5+ neurons recapitulate rostral D1-SPN activity patterns. Together, these findings establish a spatially confined rostral medNAcSh subregion as a critical regulator of reward consumption and

introduce Stard5 as a molecular tool for its manipulation, offering new opportunities for intervention in dysregulated eating.

## Introduction

Feeding behavior is a multifaceted physiological process governed by interconnected homeostatic and hedonic feeding brain circuits (*Marinescu and Labouesse, 2024*; *Stuber et al., 2025*; *Zhu et al., 2025*). While homeostatic pathways in the hypothalamus and brainstem regulate energy balance according to metabolic signals, non-homeostatic hedonic circuits are distributed more broadly throughout the brain to adjust feeding behavior according to sensory signals, motivation, associative learning, or pleasure (*Stuber et al., 2025*). Among key brain regions involved in the latter, the nucleus accumbens shell (NAcSh) has emerged as a significant player, in particular its most medial section, the medial NAcSh (medNAcSh; reviewed in *Marinescu and Labouesse, 2024*). Pharmacological studies in the 1990s and 2000s in rodents showed that inactivation of the medNAcSh dramatically increases the consumption of food rewards, even in fed animals (*Maldonado-Irizarry et al., 1995*; *Stratford and Kelley, 1997*). The medNAcSh consists of a mixed population of neurons of which a considerable proportion pause during reward consumption, an inhibitory signal that is necessary and sufficient to authorize food consumption, that is excitation suppresses, and pauses permit feeding (*Domingues et al., 2025*; *Krause et al., 2010*; *O'Connor et al., 2015*; *Pedersen et al., 2022*; *Roitman et al., 2005*; *Taha and Fields, 2006*; *Taha and Fields, 2005*). These pauses are driven primarily by cortical and limbic incoming circuits into the medNAcSh (*Reed et al., 2018*; *Vachez et al., 2021*), can be found in individual dopamine 1 receptor-positive striatal projection neurons (D1-SPNs) and D2-SPNs (*Domingues et al., 2025*; *O'Connor et al., 2015*; *Pedersen et al., 2022*), and are thought to allow rapid control over food consumption in response to motivational or sensory signals (*Kelley et al., 2005*; *O'Connor et al., 2015*). Importantly, recent studies showed that one of the primary cellular mediators of the eating-inhibitory effects in the medNAcSh are D1-SPNs, one of the two main output populations of the NAc (*O'Connor et al., 2015*; *Thoeni et al., 2020*; *Bond et al., 2020*).

Topographically, the NAcSh spans a 1–1.5 mm distance along the rostro-caudal (anterior-posterior) axis in mice, with studies demonstrating strong polarity gradients along this axis in the control of various behaviors (*Castro et al., 2016*; *Castro and Berridge, 2014a*; *Krause et al., 2010*; *Reed et al., 2018*; *Reynolds and Berridge, 2002*; *Reynolds and Berridge, 2008*; *Richard and Berridge, 2011*). For instance, microinjection of the glutamate antagonist DNQX or the GABA receptor agonist muscimol into the rostral medNAcSh promotes appetitive responses like feeding, while the same manipulations in the caudal medNAcSh trigger aversive behaviors (*Reynolds and Berridge, 2001*; *Richard and Berridge, 2011*). D1, but not D2 receptor antagonists, were able to block the glutamate-mediated feeding responses in the rostral medNAcSh, suggesting a more important role for the D1-SPN pathway (*Richard and Berridge, 2011*). Altogether, these findings suggest the existence of a hotspot for feeding located in the rostral sections of the medNAcSh, possibly involving D1-SPNs. However, the cellular, circuit, and molecular mechanisms underlying this rostral hotspot remain unknown.

Against this background, we hypothesized that D1-SPNs regulate feeding behavior according to a rostro-caudal gradient within the medNAcSh. We use fiber photometry recordings and optogenetic activation studies, finding a functional dichotomy between rostral vs. caudal medNAcSh D1-SPNs in reward consumption but not in aversive behavioral assays. We perform neuroanatomical investigations in the rostral vs. caudal medNAcSh and identify various circuit and molecular signatures of these two subregions, including a molecular marker, Stard5, densely expressed in 80% of rostral medNAcSh D1-SPNs and D2-SPNs. Finally, we generate a new Flp driver line which targets the Stard5 rostral medNAcSh subpopulation and use it in proof-of-concept experiments, finding that Stard5 cells replicate the population-level activity signatures of rostral medNAcSh D1-SPNs during reward consumption and aversion. Our work outlines the importance of functional and molecular gradients in the medNAcSh, demonstrating the existence of a D1 rostral hotspot for feeding in the medNAcSh that is partly recapitulated by Stard5 + cells, a newly described molecular marker to access this feeding hub. These findings offer new potential access points for targeting dysregulated eating through a spatially confined medNAcSh region, outside traditional homeostatic feeding circuits.

## Results

### Rostral medNAcSh D1-SPNs regulate feeding

We first set to identify points of divergence in the feeding-related properties of D1-SPNs along the rostro-caudal medNAcSh axis. It is well established that, although D1-SPNs show both excitatory and inhibitory responses during feeding, a considerable proportion of cells are inhibited in the form of prolonged pauses (*Domingues et al., 2025*; *Krause et al., 2010*; *O'Connor et al., 2015*; *Pedersen et al., 2022*; *Roitman et al., 2005*; *Taha and Fields, 2006*; *Taha and Fields, 2005*), which can be reflected at the population-level activity (*Domingues et al., 2025*). These pauses are thought to authorize eating by disinhibiting downstream circuits (*O'Connor et al., 2015*; *Thoeni et al., 2020*; *Bond et al., 2020*). To determine whether such pause responses change along the rostro-caudal axis, we used fiber photometry to measure population-level D1-SPN activity in both the rostral and caudal medNAcSh while sated mice were fed a palatable food source (condensed milk). Drd1-cre mice received an adeno-associated virus (AAV) expressing a cre-dependent calcium indicator (jGCaMP8m *Zhang et al., 2023b*) and an optic fiber above the rostral or caudal medNAcSh, targeting specifically the dorsal region given its strong role in feeding (*O'Connor et al., 2015*; *Figure 1a–b*; see also *Figure 1—figure supplement 1*). Mice were allowed to consume up to 30 food rewards per session delivered at random intervals (*Figure 1c*). As expected, population D1-SPN activity in the rostral medNAcSh showed a prolonged pause upon reward consumption, with average activity decreasing at eating onset and returning to baseline after the 10 s consumption period (*Figure 1d*). This was reflected by a significant decrease in the $\Delta F/F0$ minima during the consumption as compared to the pre-consumption period (*Figure 1e*). The inhibition of D1-SPN activity during reward consumption in rostral medNAcSh aligns well with previous findings (*Bond et al., 2020*; *Krause et al., 2010*; *O'Connor et al., 2015*; *Thoeni et al., 2020*). On the other hand, D1-SPNs in the caudal medNAcSh showed an acute excitatory response at consumption onset that returned to baseline after ~2.5 s, followed by a minor prolonged inhibition (*Figure 1f*). This was reflected by a significant increase in the $\Delta F/F0$ maxima in the consumption as compared to the pre-consumption period (*Figure 1g*). To confirm regional specificity of the recordings, histological validation in a subset of animals showed correct placement of the optic fibers at the intended stereotaxic coordinates. Additionally, injections targeting the rostral medNAcSh exhibited minimal to no viral expression at caudal coordinates, while injections targeting the caudal medNAcSh showed minimal to no viral expression at rostral coordinates (*Figure 1—figure supplement 3*). These data indicate a strong divergence in the responses of D1-SPNs to reward consumption depending on their topographical location, highlighting the existence of a rostro-caudal gradient. It further demonstrates that the rostral, but not caudal, medNAcSh represents the primary locus for the expression of pause responses by D1-SPNs at a population level during reward consumption.

Previous work showed that medNAcSh D1-SPNs can also causally impact hedonic eating, where optogenetic activation of D1-SPNs (using rostral-centric medNAcSh coordinates) inhibits reward consumption, while optogenetic inhibition promotes consumption independent of metabolic needs (*Bond et al., 2020*; *O'Connor et al., 2015*). This is similar to what is found with global, cell-unspecific, medNAcSh pharmacological inhibition (*Maldonado-Irizarry et al., 1995*; *Stratford and Kelley, 1997*). Based on this extensive literature and our photometry results, we asked whether reward consumption effects were confined to the rostral side of the medNAcSh or whether similar functions extended to the caudal medNAcSh. Drd1-cre mice received bilateral injections of an AAV expressing a cre-dependent excitatory opsin, ChrimsonR, and optic fibers above the rostral or caudal medNAcSh (*Figure 2a-b*, *Figure 1—figure supplement 3e-h*). We first conducted a proof-of-concept experiment in a small cohort to validate previously published optogenetic protocols. Mice were exposed to palatable food rewards with alternating blocks of 8 min: blocks 1 and 3 without laser stimulation (laser off) and block 2 with stimulation (laser on; *Figure 2c*). In ChrimsonR Rostral medNAcSh mice, lick count was initially high with laser off, then decreased during laser on, and rebounded again with laser off. This consumption pattern is very similar to what is seen in the literature (*Bond et al., 2020*; *O'Connor et al., 2015*). When averaging lick counts per epoch type, we could detect a significant reduction in reward consumption with optogenetic stimulation of rostral medNAcSh D1-SPNs. No significant change in consumption was seen in mCherry controls. Although these mice began with a slightly lower (non-significant) baseline compared to ChrimsonR mice, there was no evidence of a floor effect (*Figure 2d–e*). Altogether, this feasibility experiment confirmed published findings (*Bond et al.,*

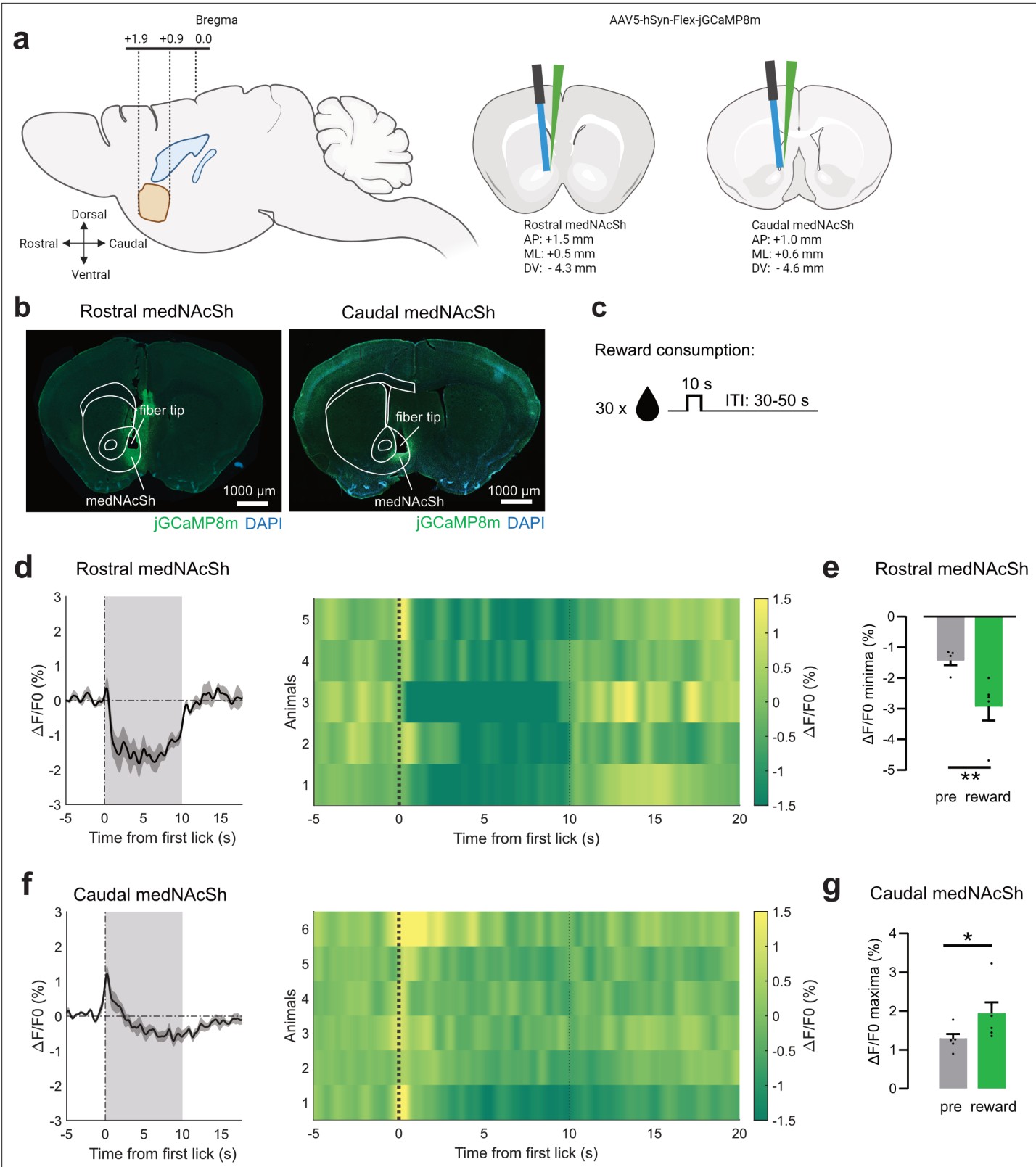

**Figure 1.** D1-SPNs in the rostral vs. caudal medNAcSh respond differentially to reward consumption. (**a**) Surgery schematics of Drd1-cre mice injected with a cre-dependent calcium indicator (jGCaMP8m) and implanted with unilateral optic fibers in the rostral or caudal medNAcSh to image calcium photometry signals. Stereotaxic coordinates for injection and optic fiber implantation in the rostral and caudal are shown on the right. (**b**) Representative coronal images of rostral (left) vs. caudal (right) medNAcSh mice. (**c**) Protocol for the unpredicted reward consumption behavioral task: 30 rewards

*Figure 1 continued on next page*

*Figure 1 continued*

were available for 10 s at random intervals. (**d**) Left: Average calcium activity in D1-SPNs (all trials) shown as normalized fluorescence (ΔF/F0, %) in the rostral medNAcSh over time aligned to the onset of reward consumption, that is first lick onset (0 s). Grey shading represents the 10 s epoch for reward consumption. Right: Heatmap showing average calcium activity across individual trials (each row represents an animal) aligned to reward consumption, i.e. lick onset (0 s, thick dotted line). Thin dotted line: end of reward access. On average, mice completed 16.2 of 30 trials successfully (54%). (**e**) Quantification of the data in (**d**) depicts a significant decrease in rostral medNAcSh activity upon reward consumption, as shown by a significant decrease in ΔF/F0 minima in the 0–5 s reward epoch vs. –6 to –1 s pre-reward epoch. Paired t-test, t(4) = 4.610, p=0.0100. N=5 mice, 1 female, 4 male. (**f**) same as (**d**) for the caudal medNAcSh. Animals completed an average of 17.67 successful trials out of 30 (59%). (**g**) Quantification of the data in (**f**) depicts a significant increase in caudal medNAcSh activity upon reward consumption, as shown by a significant increase in ΔF/F0 maxima in the 0–5 s reward epoch vs. –6 to –1 s pre-reward epoch. Paired t-test, t(5) = 3.059, p=0.0281. N=6 mice, 4 female, 2 male. Data is mean ± SEM. *p<0.05; **p<0.01; ***p<0.001. Panel a was created with BioRender. See also *Figure 1—figure supplement 1*.

The online version of this article includes the following figure supplement(s) for figure 1:

**Figure supplement 1.** Projections sites of D1-SPNs from the rostral and caudal medNAcSh.

**Figure supplement 2.** Learning rate (% successful trials) during reward consumption in animals undergoing fiber photometry recordings.

**Figure supplement 3.** Representative images of rostral and caudal optic fiber placements and viral spread.

---

*2020*; *O'Connor et al., 2015*), namely that activation of rostral medNAcSh D1-SPNs inhibits reward consumption. We then went on to test a full experimental cohort to evaluate the differential contribution of rostral vs. caudal medNAcSh D1-SPNs in reward consumption. Here, we modified the protocol to enhance the sensitivity of the assay, minimize potential confounds such as increasing satiety over time, and improve our ability to detect even minor differences in reward consumption between the three groups. We started with a laser on epoch and expanded the number of epochs from 3 to 5 (3 laser on, 2 laser off); we also reduced epoch duration from 8 to 5 min (*Figure 2f*). As before, and despite the modified protocol, we again found that optogenetic stimulation of D1-SPNs in the rostral medNAcSh significantly inhibited reward consumption with no effects in mCherry controls (*Figure 2g–i*). A similar (as seen in *Figure 2d*) rebound in consumption was detected after laser on epochs in rostral medNAcSh mice but not mCherry controls (*Figure 2g*). Importantly, although there was a non-significant trend for an opto-mediated eating-inhibitory effect in caudal medNAcSh mice (see *Figure 2g*), and the number of caudal medNAcSh animals demonstrating opto-mediated inhibition of consumption was 64% (vs. 100% of rostral medNAcSh mice and 30% mCherry; *Figure 2i*), the mean effect was too small to reach statistical significance (p=0.5244) (*Figure 2h*). To determine whether the reduced reward consumption observed in Rostral ChrimsonR mice could be explained by changes in locomotion, we quantified the total distance traveled during this task. Optogenetic stimulation led to an increase in locomotion in the small cohort of Rostral ChrimsonR mice in the reward consumption experiment shown in *Figure 2d–e* (see *Figure 2—figure supplement 2a*), while no change in locomotion was observed across epochs in mCherry controls, ChrimsonR Rostral, and Caudal mice (*Figure 2—figure supplement 2b*, related to *Figure 2g–i*). This indicates that the reduced feeding in Rostral ChrimsonR mice is not due to impaired locomotion. In sum, rostral, but not caudal, medNAcSh D1-SPNs inhibit reward consumption. Altogether, this indicates the existence of a rostro-caudal gradient in the regulation of hedonic eating by medNAcSh D1-SPNs, supporting the existence of a D1 rostral medNAcSh appetitive hotspot.

## Rostral and caudal medNAcSh D1-SPNs show similar function in aversive assays

Previous work has shown that the medNAcSh plays strong roles in not only appetitive but also aversive behaviors (*Al-Hasani et al., 2015*; *de Jong et al., 2019*; *Domingues et al., 2025*; *Liu et al., 2022*; *Yang et al., 2018*; *Zhou et al., 2022*). For instance, Liu et al. showed that medNAcSh D1 projections to the ventral pallidum (VP; but not D1 projections to the ventral tegmental area, VTA) are acutely activated by aversive stimuli and in turn optogenetic stimulation of these projections promotes aversion in a real time place preference vs. avoidance assay (RTTPA; *Liu et al., 2022*). We therefore asked whether regulation of aversive behaviors by medNAcSh D1-SPNs differed along the rostro-caudal axis, as was the case for the reward consumption task (*Figures 1 and 2*). Using fiber photometry, we first recorded D1-SPN population activity in the rostral and caudal medNAcSh while mice were exposed to aversive stimuli in the form of 10 unpredicted shocks delivered at random intervals (*Figure 3a and b*). We found that both rostral and caudal D1-SPNs showed similar responses (albeit not fully overlapping

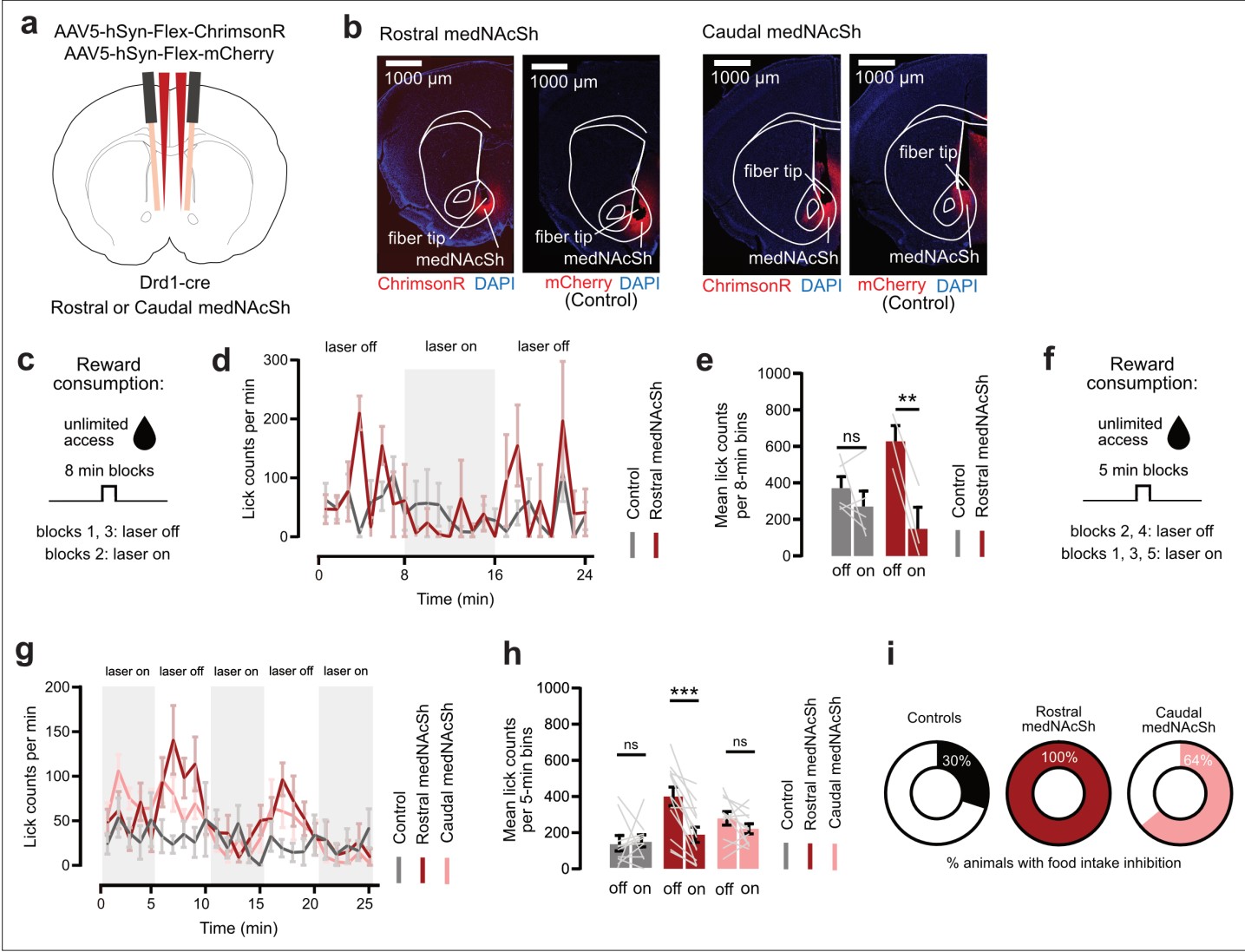

**Figure 2.** In vivo optogenetic stimulation of rostral medNAcSh inhibits reward consumption. (**a**) Surgery schematics of Drd1-cre mice injected with a cre-dependent activating opsin (ChrimsonR) or mCherry control AAV and implanted with bilateral optic fibers in the rostral or caudal medNAcSh to stimulate local D1-SPNs. (**b**) Representative coronal images of rostral (left) vs. caudal (right) medNAcSh mice expressing relevant AAVs. (**c**) Protocol for the proof-of-concept reward consumption experiment: mice had unlimited access to the reward spout. Blocks of 8 min without (laser off, 'non-opto') or with opto-stimulation (laser on, 'opto') were alternated (off/on/off) for a total of three blocks. (**d**) Quantification of the lick counts per min in mCherry mice vs. ChrimsonR (rostral medNAcSh) mice, showing a lower lick count per min in rostral medNAcSh mice during opto stimulation vs. 'non-opto' epochs; lick counts unchanged across epochs in mCherry mice. (**e**) Quantification of mean licks per session in the opto-stimulation vs. non-opto-stimulation epochs shows a significant decrease in lick counts following stimulation of rostral medNAcSh D1-SPNs (and not in mCherry controls) confirming published studies. Two-way RM-ANOVA (group x epoch). Main effects: group $F_{(1,6)} = 0.4089$, p=0.5461; epoch $F_{(1,6)} = 18.44$, p=0.0051; group x epoch $F_{(1,6)} = 7.846$, p=0.0311. Sidak post-hoc opto-stimulation vs. non-opto-stimulation: Controls $t_{(6)} = 1.219$, p=0.4648; ChrimsonR Rostral medNAcSh $t_{(6)} = 4.488$, p=0.0083; N=5 Controls, 3 female, 2 male; N=3 ChrimsonR Rostral medNAcSh, 1 female, 2 male. (**f**) Protocol for the full reward consumption experiment: mice had unlimited access to the reward spout. Blocks of 5 min with or without opto-stimulation were alternated (on/off/on/off/on) for a total of five blocks. (**g**) Quantification of the lick counts per min, showing that ChrimsonR rostral medNAcSh mice eat less during on vs. off epochs as in (**d**); there is a trend for a similar observation for ChrimsonR caudal medNAcSh mice. No change across epochs for mCherry control mice, as in (**d**). (**h**) Quantification of mean licks per session in the opto-stimulation vs. non-opto-stimulation epochs shows a significant decrease in lick counts following stimulation of rostral, but not caudal medNAcSh D1-SPNs (and no change in mCherry controls), indicating that activation of rostral but not caudal medNAcSh D1-SPNs significantly decreases reward consumption. Two-way RM-ANOVA (group x epoch). Main effects: group $F_{(2,31)} = 4.548$, p=0.0185; epoch $F_{(1,31)} = 10.58$, p=0.0028; group x epoch $F_{(2,31)} = 6.651$, p=0.0039. Sidak post-hoc opto-stimulation vs. non-opto-stimulation: Controls $t_{(31)} = 0.2669$, p=0.9909; Rostral medNAcSh $t_{(31)} = 5.010$, p<0.0001; Caudal medNAcSh $t_{(31)} = 1.253$, p=0.5244. N=10 Controls, 5 female, 5 male; N=13 Rostral medNAcSh, 6 female, 7 male; N=11 Caudal medNAcSh, 4 female, 7 male. (**i**) Pie charts showing % of mice showing food intake inhibition (mean Δlick counts non-opto/opto >0) in each group: 100% of ChrimsonR rostral medNAcSh mice, 30% of controls; and although caudal medNAcSh opto-

*Figure 2 continued on next page*

Figure 2 continued

stimulation does not lead to significant reward consumption inhibition as shown in (**h**), 64% of mice show an inhibition, indicating the effect is present but the effect size is too small to get captured statistically. Data is mean ± SEM. *p<0.05; **p<0.01; ***p<0.001. See also *Figure 2—figure supplement 1*.

The online version of this article includes the following figure supplement(s) for figure 2:

**Figure supplement 1.** Additional analyses of in vivo optogenetic stimulation of medNAcSh cells during reward consumption.

**Figure supplement 2.** Optogenetic stimulation of rostral medNAcSh inhibits reward consumption and promotes aversion without reducing locomotion.

in the period post-shock), with both groups depicting strong acute excitatory responses to the shocks (*Figure 3c and e*). This was reflected by a significant increase in the ΔF/F0 in the shock vs. pre-shock period in both groups (*Figure 3d and f*). Given prior reports of cell-type differences in NAcSh responses to aversive stimuli (*Liu et al., 2022*) and the absence of differences detected in our assay, our findings likely reflect a common activation of rostral and caudal D1-SPNs during acute aversive events. To further probe for potential regional differences, confirm the sensitivity of our approach, and to ensure that this effect is not specific to a single stressor, we also assessed responses to a distinct aversive stimulus: tail lift, a recognized ethologically relevant stressor (*Figure 3g*). Here again, we identified similar acute excitatory responses in rostral and caudal D1-SPNs to the stimuli, reflected by significant increases in ΔF/F0 in the lift vs. pre-lift period (*Figure 3h–k*). Hence, at the population level, rostral and caudal medNAcSh D1-SPNs similarly respond to aversive stimuli. Finally, we determined whether direct optogenetic manipulation of these neurons would trigger aversive responses in a well-established RTPPA assay (*Al-Hasani et al., 2015*; *Liu et al., 2022*; *Zhou et al., 2022*; *Figure 4a–d*). We found that optogenetic stimulation of both rostral and caudal medNAcSh D1-SPNs promoted avoidance of the opto-paired chamber, with no effects in controls, suggesting that stimulation of these neurons is aversive. Quantification of locomotion showed no reduction in distance traveled in the light-paired chamber (*Figure 2—figure supplement 2c*), indicating that the avoidance was not driven by impaired locomotion. These data indicate that medNAcSh D1-SPNs generally promote aversion without affecting locomotion and without major differences along the rostro-caudal axis. This lack of functional differences between the rostral and caudal medNAcSh contrasts with other studies showing divergent responses between, for example, medial and lateral NAcSh (*Chen et al., 2023*) or medNAcSh D1-SPN projections to different downstream regions (*Liu et al., 2022*). Altogether, our data highlight the existence of a rostral D1 medNAcSh hotspot that is primarily restricted to an appetitive feeding context.

## Molecular rostro-caudal gradients across the medNAcSh highlight *Stard5* and *Peg10* as unique markers

Rostro-caudal functional differences in feeding could reflect distinct molecular compositions across the medNAcSh. Indeed, recent work using fluorescent in-situ hybridization (FISH) or single-cell RNA sequencing (scRNAseq) has delineated the strong molecular and spatial diversity within the NAc (*Chen et al., 2021*; *Stanley et al., 2020*), identifying cellular clusters within D1-SPNs and/or D2-SPNs that express combinations of marker genes and are often enriched in spatial subregions of the NAc. Hence it is possible that unique molecular markers are enriched in the rostral vs. caudal medNAcSh and contribute to functional differences. We used published in-situ hybridization datasets from the Allen Mouse Brain Atlas (*Lein et al., 2007*) to identify novel molecular markers enriched in one of these two subregions. Screening 48 markers flagged by the Allen Brain Atlas as expressing in the NAc, we found that most genes showed relatively stable expression across the rostro-caudal axis (e.g. *Rasal1, LOC381076* [now called *Baiap3*]*, Stra6, Lypd1*; which are representative examples shown in *Figure 5—figure supplement 1*). However, we detected five genes depicting differential rostro-caudal gradient expression: *Stard5, Smug1*, and *Cartpt* were enriched in the rostral medNAcSh while *Peg10* and *Sgsm1* were enriched in the caudal medNAcSh (*Figure 5a*), which was also confirmed by optical density analysis (*Figure 5b*). Among all markers, *Stard5* and *Peg10* showed particular strong enrichment in the rostral and caudal medNAcSh, respectively, as opposed to neighboring regions (*Figure 5b*). These data demonstrate the existence of molecular gradients across the rostro-caudal axis of the medNAcSh and highlight *Stard5* and *Peg10* as markers enriched in rostral vs. caudal medNAcSh, respectively.

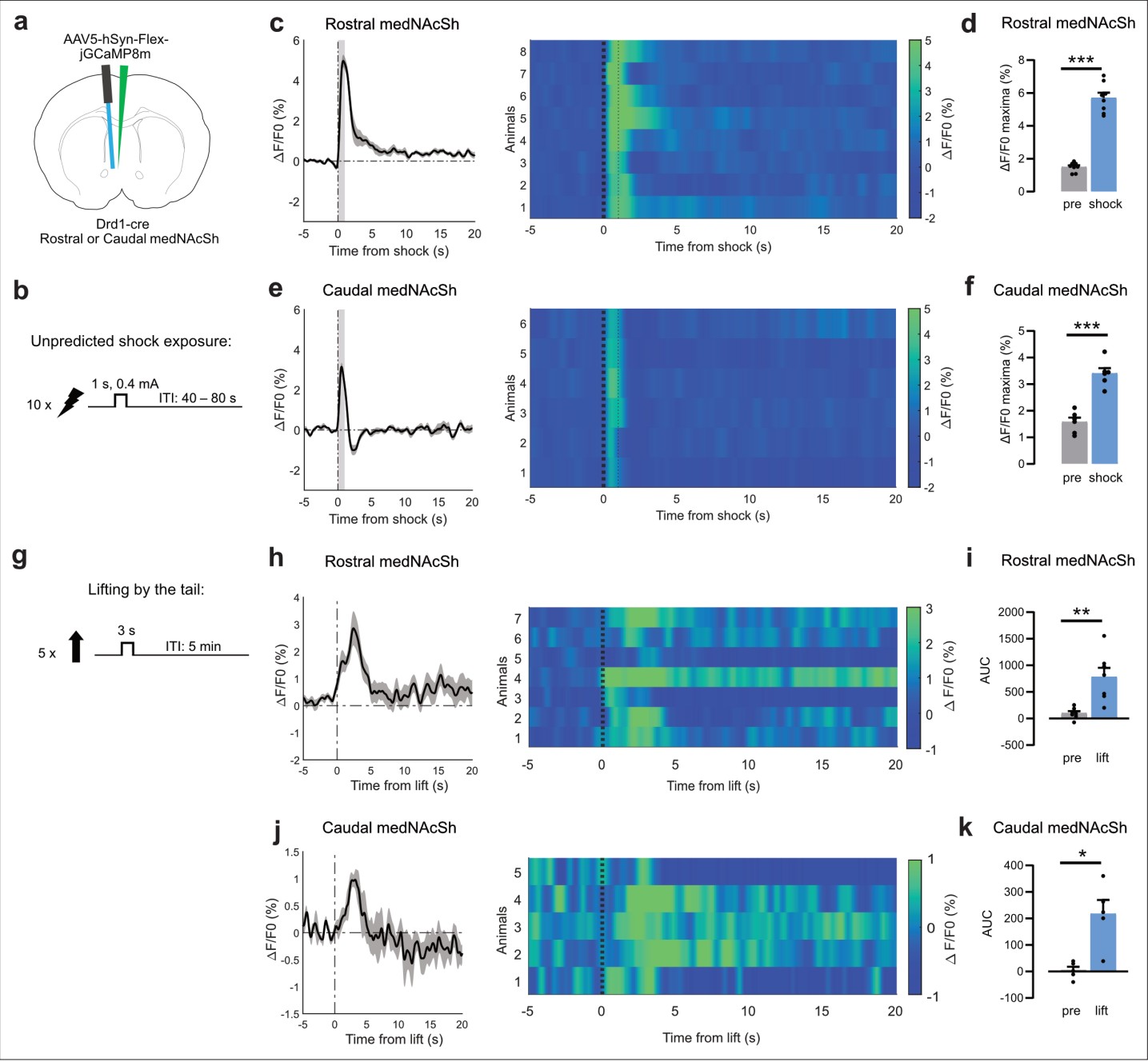

**Figure 3.** D1-SPNs in the rostral vs. caudal medNAcSh respond similarly to aversion. (**a**) Surgery schematics of Drd1-cre mice injected with a cre-dependent calcium indicator (jGCaMP8m) and implanted with unilateral optic fibers in the rostral or caudal medNAcSh to image calcium photometry signals. (**b**) Protocol for the unpredicted shock behavioral task: 10 shocks (0.4 mA) were available for 1 s at random intervals. (**c**) Left: Average calcium activity in D1-SPNs (all trials) shown as normalized fluorescence (ΔF/F0, %) in the rostral medNAcSh over time aligned to shock onset (0 s), showing acute excitation at shock onset, followed by a slow return to baseline. Grey shading: shock epoch. Right: Heatmap showing calcium activity across individual trials (each row represents a trial) aligned to shock onset (0 s, thick dotted line). Thin dotted line: end of shock. (**d**) Quantification of the data in (**c**) depicts a significant increase in rostral medNAcSh activity upon shock onset, as shown by a significant increase in ΔF/F0 maxima in the 0–5 s shock epoch vs. –6 to –1 s pre-shock epoch. Paired t-test, t(7) = 14.91, p=0.0001. N=8 mice, 2 female, 6 male. (**e**) Same as (**c**) for the caudal medNAcSh showing acute excitation at shock onset, followed by a rapid return below zero before returning to baseline; the significance of this is unknown. (**f**) Same as (**d**) for the caudal medNAcSh. Paired t-test, t(5) = 8.312, p=0.0004. N=6 mice, 4 female, 2 male. (**g**) Protocol for the tail lift task: 5 tail lifts (3 s to lift the mouse, followed by 55 s stabilized in the air then back to the cage. Only the first 20 s are shown), at 5 min intervals. (**h**) Left: Average calcium activity in D1-SPNs (all trials) shown as normalized fluorescence (ΔF/F0, %) in the rostral medNAcSh over time aligned to lift onset (0 s), showing acute excitation at lift onset, followed by return to baseline. Right: Heatmap showing calcium activity across individual trials (each row represents a trial) aligned to lift

*Figure 3 continued on next page*

*Figure 3 continued*

onset (0 s, thick dotted line). (**i**) Quantification of the data in (**h**) depicts a significant increase in rostral medNAcSh activity upon lift onset, as shown by a significant increase in ΔF/F0 maxima in the 0–4 s lift epoch vs. –4–0 s pre-lift epoch. Paired t-test, t(6) = 4.214, p=0.0056. N=7 mice, 2 female, 5 male. (**j**) Same as (**h**) for the caudal medNAcSh showing acute excitation at lift onset (starts at t=0 s), followed by a rapid return below zero before returning to baseline; the significance of this is unknown. (**k**) Same as (**i**) for the caudal medNAcSh. Paired t-test, t(4) = 4.435, p=0.0114. N=5 mice, 3 female, 2 male. Data is mean ± SEM. ns non-significant; *p<0.05; **p<0.01; ***p<0.001.

We analyzed a published scRNAseq dataset of the entire NAc (core and shell; *Chen et al., 2021*) to further characterize the cellular expression patterns of *Stard5* and *Peg10* (*Figure 6a–c*, see also below). We found that, in the NAc, *Stard5* is expressed predominantly in D1-SPNs (34.91% of Stard5 + cells) and D2-SPNs (30.41%) with lesser expression in other cell types, mostly interneurons (17.60%; *Figure 6—figure supplement 1a*). A similar pattern was observed for *Peg10*, with strongest expression in D1-SPNs (36.52% of Peg10 + cells), D2-SPNs (26.69%), and interneurons (33.13%; *Figure 6— figure supplement 1b*).

Since *Stard5* and *Peg10* are expressed in both D1- and D2-SPNs, we performed triple FISH to further characterize their expression profiles within these two main cell populations (D1- and D2-SPNs) across subregions of the NAc (*Figure 6d and f*). First, we confirmed that *Stard5* is strongly expressed within the D1-SPN (30.62% of *Drd1*+ cells) and D2-SPN (32.47% of *Drd2*+ cells) populations amongst the entire NAc, similar to what we see with scRNAseq 'entire NAc' data (29.55% of *Drd1*+ cells and 38.19% of *Drd2*+ cells; *Figure 6b*). Importantly, subregion analyses revealed that *Stard5* is predominantly expressed in the rostral and intermediate sections of the medNAcSh, in particular its dorsal part, precisely the rostral feeding hotspot identified in this study and others (*Reed et al., 2018*; *Reynolds and Berridge, 2001*; *Richard and Berridge, 2011*; *Figure 6e*). *Stard5* was expressed in a

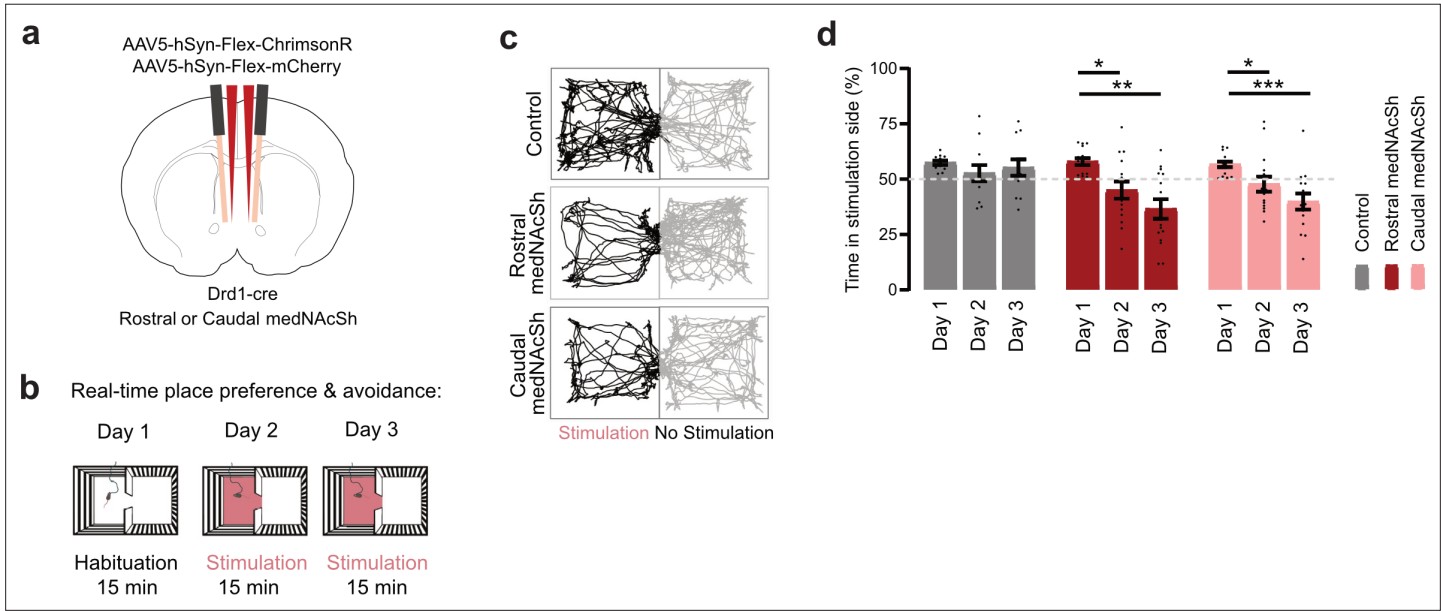

**Figure 4.** Optogenetic stimulation of D1-SPNs in the rostral and caudal medNAcSh lead to aversion. (**a**) Surgery schematics of Drd1-cre mice injected with a cre-dependent activating opsin (ChrimsonR) or mCherry control AAV and implanted with bilateral optic fibers in the rostral or caudal medNAcSh to stimulate local D1-SPNs. (**b**) Protocol for the real-time place preference or avoidance (RTPPA) behavioral task: After 1 habituation (unstimulated) day, mice were tested on 2 days where opto-stimulation occurred as soon as mice entered the laser-paired chamber (session duration 15 min). (**c**) Representative locomotor patterns of animals in the RTPPA chamber on day 3 in all groups. (**d**) Quantification of the percent (%) time spent in the stimulation side vs. non-stimulated side. Opto-stimulation led to a decrease in the time spent in the stimulation chamber in rostral medNAcSh and caudal medNAcSh mice but not in mCherry mice, indicating that activation of rostral and caudal medNAcSh D1-SPNs is aversive. Two-way RM-ANOVA (group x day). Main effects: group F(2,39) = 3.539, p=0.0387; day F(1.822,71.05)=19.08, p=0.0001; group x day F(4,78) = 3.302, p=0.0149. Sidak post-hoc day 2 vs. day 1: Controls t(11) = 1.232, p=0.4277; Rostral medNAcSh t(14) = 2.951, p=0.0209; Caudal medNAcSh t(14) = 2.700, p=0.0342. Sidak post-hoc day 3 vs. day 1: Controls t(11) = 0.5776, p=0.8195; Rostral medNAcSh t(14) = 4.250, p=0.0016; Caudal medNAcSh t(14) = 4.661, p=0.0007. N=12 Controls, 7 female, 5 male; N=15 Rostral medNAcSh, 7 female, 8 male; N=15 Caudal medNAcSh, 7 female, 8 male. Data is mean ± SEM. ns non-significant; *p<0.05; **p<0.01; ***p<0.001.

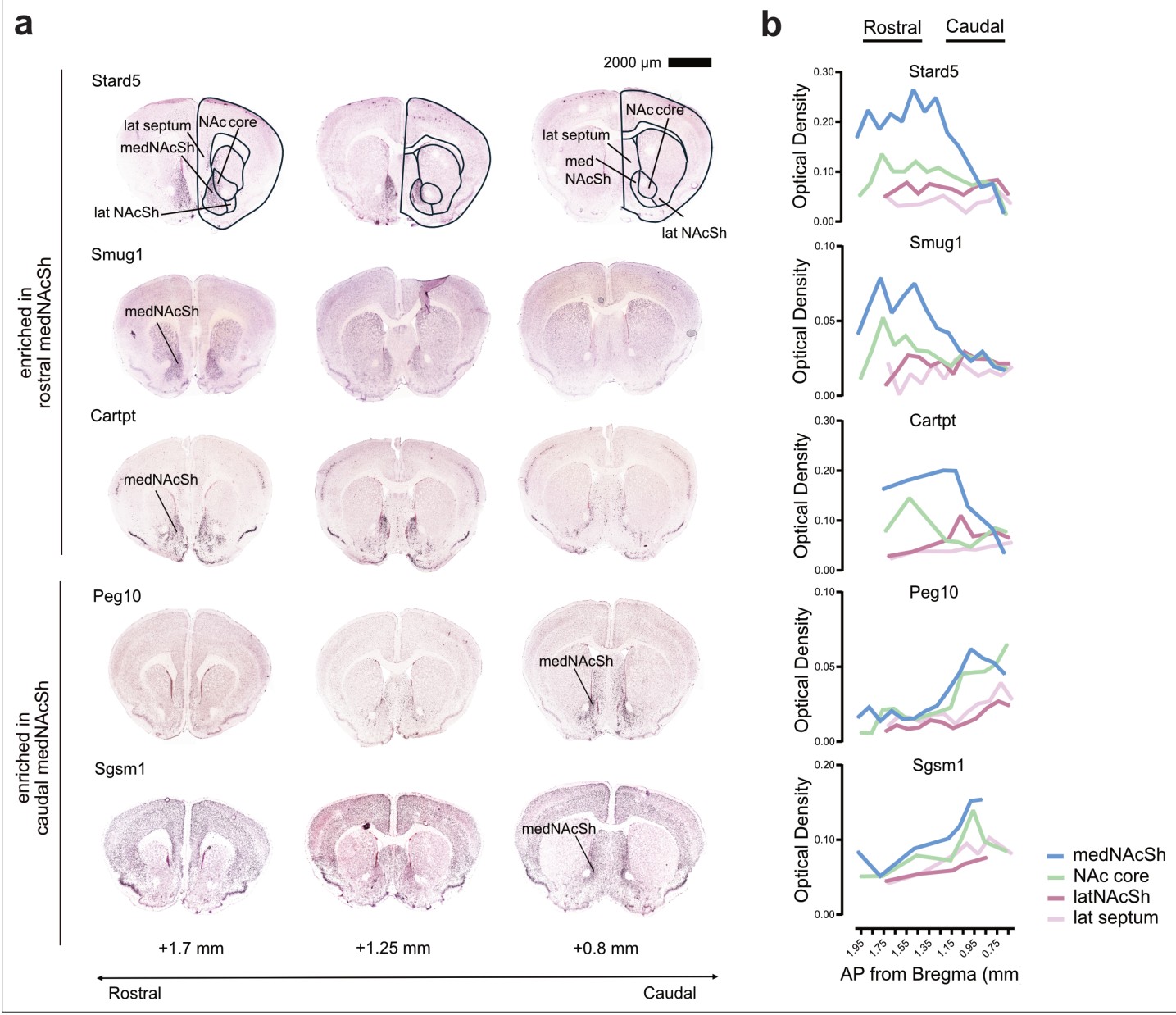

**Figure 5.** *Stard5* is enriched in the rostral region and *Peg10* in the caudal region of the medNAcSh, respectively. (**a**) Representative in-situ hybridization images obtained from the Allen Brain Atlas (https://mouse.brain-map.org/) for five markers showing enriched expression in the rostral medNAcSh (*Stard5, Smug1, Cartpt*) or the caudal medNAcSh (*Peg10, Sgsm1*). Data for the NAc core, lateral NAcSh (latNAcSh), and lateral septum (lat septum) are also shown. All 48 markers available in the Allen Brain Atlas labeling the nucleus accumbens were screened. Images on the left are more rostral (bregma +1.65 mm); images on the right are more caudal (bregma +0.85 mm). (**b**) Quantification of expression levels of markers from (**a**) in the medNAcSh along the rostral-caudal axis (bregma +1.95 mm to +0.75 mm) using optical density measurements (n=1 mouse, 7–14 sections total, data shown is the mean from 2 measures/section). *Stard5, Smug1,* and *Cartpt* show enrichment in the rostral medNAcSh, with *Stard5* showing the strongest total expression and strongest relative expression in medNAcSh as compared to other regions. *Peg10* and *Sgsm1* show enrichment in the caudal medNAcSh, with *Peg10* showing the strongest relative expression in medNAcSh as compared to latNAcSh and lateral (lat) septum (but not NAc core). See also **Figure 5—figure supplement 1**.

The online version of this article includes the following figure supplement(s) for figure 5:

**Figure supplement 1.** Molecular markers for the medNAcSh.

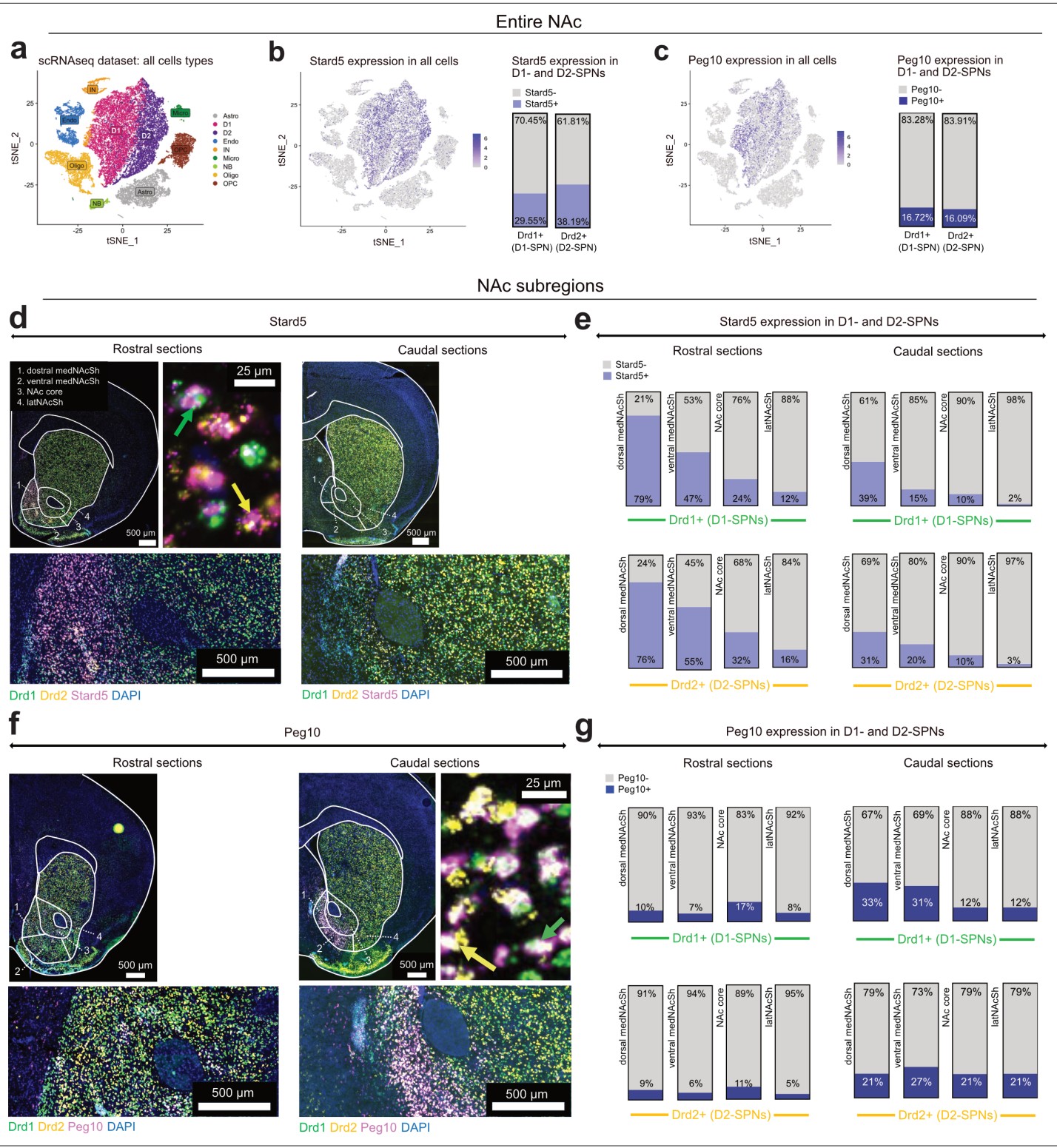

**Figure 6.** *Stard5* is enriched in D1- and D2-SPNs of the rostral medNAcSh. (**a**) t-SNE plot showing cell types in the entire NAc based on re-analysis of a published scRNAseq dataset, GSE118020 (***Chen et al., 2021***). Data is pooled from 11 mice, 36,670 cells. Different cell clusters are color-coded. Clusters mark cell types identified by canonical markers, as used in ***Chen et al., 2021***. Astro: astrocytes, D1: D1-SPN, D2: D2-SPN, Endo: endothelial cells, IN: interneurons, Micro: microglia, NB: neuroblasts, Oligo: oligodendrocytes, OPC: oligodendrocyte progenitor cells. (**b**) Left: t-SNE plots showing expression of Stard5 across NAc cells; the densest expression is found in the D1- and D2-SPNs clusters. *Stard5* expression level is color-coded from 0 (low) to 6 (high). Right: Quantification of cells positive or negative for *Stard5* within D1- and D2-SPNs (identified by *Drd1* and *Drd2* expression). About

*Figure 6 continued on next page*

Figure 6 continued

1/3 of D1- and 1/3 of D2-SPNs express *Stard5*. (**c**) Left: t-SNE plots showing expression of *Peg10* across NAc cells; the densest expression is found in the D1-SPN, D2-SPN, and interneuron clusters. *Peg10* expression level is color-coded from 0 (low) to 6 (high). Right: Quantification of cells positive or negative for *Peg10* within D1- and D2-SPNs (identified by *Drd1* and *Drd2* expression). A small proportion (16 to 17%) of D1- and of D2-SPNs express *Peg10*. (**d**) Left: Representative image of a fluorescent in-situ hybridization (RNAscope) assay for *Drd1*, *Drd2* (labeling D1- and D2-SPNs), and *Stard5* in subregions of the rostral and caudal sections of the NAc (dorsal and ventral medNAcSh, NAc core and latNAcSh). Lower panel: zoom-in and Right panel: high-resolution magnification showing *Stard5+*, *Drd1+* (green arrow) or *Stard5+*, *Drd2+* (yellow arrow) neurons. (**e**) Quantification of expression of *Stard5* in D1- and D2-SPNs in subsections of the NAc (dorsal and ventral medial [med] NAcSh, NAc core and lateral [lat] NAcSh) in the rostral and caudal regions of the NAc. An average of 6–8 sections in total, including 2–3 for most rostral sections and 2–3 for most caudal sections; and 5 mice in total were analyzed. Data shows strong enrichment (75–80%) of *Stard5* in D1- and D2-SPNs of the rostral, dorsal medNAcSh (and much less in the caudal medNAcSh or other NAc regions). See also **Figure 6—figure supplement 1** showing minor *Stard5* expression in other cell types. (**f**) Same as (**d**) for *Peg10*. (**g**) Same as (**e**) for *Peg10* (n=5 mice). Data shows enrichment of *Peg10* in D1- and D2-SPNs of the dorsal and ventral caudal medNAcSh (very little expression in the rostral medNAcSh). See also **Figure 6—figure supplement 1** showing *Peg10* expression in interneurons (in addition to D1- and D2-SPNs).

The online version of this article includes the following figure supplement(s) for figure 6:

**Figure supplement 1.** *Stard5* and *Peg10* expression patterns in the NAc.

large majority of D1-SPNs (79% of *Drd1+* cells) and D2-SPNs (76% of *Drd2+* cells) in the dorsal rostral medNAcSh (**Figure 6e**), hence representing an almost ubiquitous marker for SPNs in this region. *Stard5* expression in D1-SPNs and D2-SPNs gradually decreased when moving more caudally, ventrally, and laterally into the NAc core and lateral shell (**Figure 6e**), confirming that its expression follows a molecular gradient rather than a binary on/off expression. We performed the same FISH analysis for *Peg10*, and found a similar gradient pattern with strongest expression in the caudal medNAcSh and gradually decreasing expression moving more rostrally, ventrally, and laterally (**Figure 6f and g**). In the caudal medNAcSh, *Peg10* was expressed in 16.45% of D1-SPNs (*Drd1+*) and 13.34% of D2-SPNs (*Drd2+*), similar to what we see with scRNAseq 'entire NAc' data (16.72% of *Drd1+* cells and 16.09% of *Drd2+* cells; **Figure 6c**). Moreover, in caudal sections, *Peg10* was enriched in dorsal and ventral medNAcSh D1-SPNs (33% dorsal and 31% ventral, of *Drd1+* cells) similar to *Stard5*; while it showed relatively stable expression across NAc subregions for D2-SPNs (i.e. no medNAcSh enrichment; **Figure 6g**). In summary, we confirm the existence of molecular gradients across the rostro-caudal medNAcSh axis, revealing *Stard5* as a strong marker for both types of SPNs in the rostral medNAcSh feeding hotspot, in particular its dorsal part.

## Medial NAcSh Stard5 cells recapitulate phenotypes of rostral medNAcSh D1-SPNs

*Stard5* could represent an interesting molecular marker to label or functionally target cellular populations in the rostral medNAcSh feeding hotspot. To evaluate this possibility, we used CRISPR/Cas9 genetic engineering to generate a new driver line, Stard5-Flp, expressing the Flp recombinase in *Stard5+* cells (**Figure 7a**, **Figure 7—figure supplement 1**), that is primarily in *Stard5+* D1-, D2-SPNs, and interneurons. Stard5-Flp-positive mice bred well, were viable, and showed no behavioral or metabolic abnormalities as compared to Stard5-Flp-negative mice indicating absence of obvious genomic off-target expression (**Figure 7—figure supplement 2**). Stard5-Flp mice were injected with an AAV expressing a Flp-dependent fluorophore (jGCaMP8m) into the rostral medNAcSh or in neighboring regions (**Figure 7b**). We found strong fluorophore expression in the rostral medNAcSh but very low or no expression when injecting in the neighboring NAc core and dorsal striatum, respectively (**Figure 7c**), confirming that *Stard5* expression is enriched in the medNAcSh as opposed to neighboring regions (**Figure 6**). Very low ectopic expression was found in Stard5-Flp-negative mice (a ratio 1:16 vs Stard5-Flp-positive mice, **Figure 7c**), confirming minimal (but not entirely absent) Flp leakage of the AAV construct, a very common yet often ignored issue, which could be mitigated in the future using modern AAV design strategies (**Fischer et al., 2019**).

Finally, we used the Stard5-Flp mouse to determine whether Stard5 cells recapitulate some of the feeding-related phenotypes of rostral medNAcSh D1-SPNs. Stard5-Flp mice were injected in the rostral medNAcSh with an AAV expressing a synaptophysin tagged Flp-dependent EGFP to evaluate the projection sites of rostral medNAcSh Stard5 + cells. We found strong expression in the VP, VTA, and lateral hypothalamus (LH; **Figure 7—figure supplement 3**), confirming that Stard5 + cells project

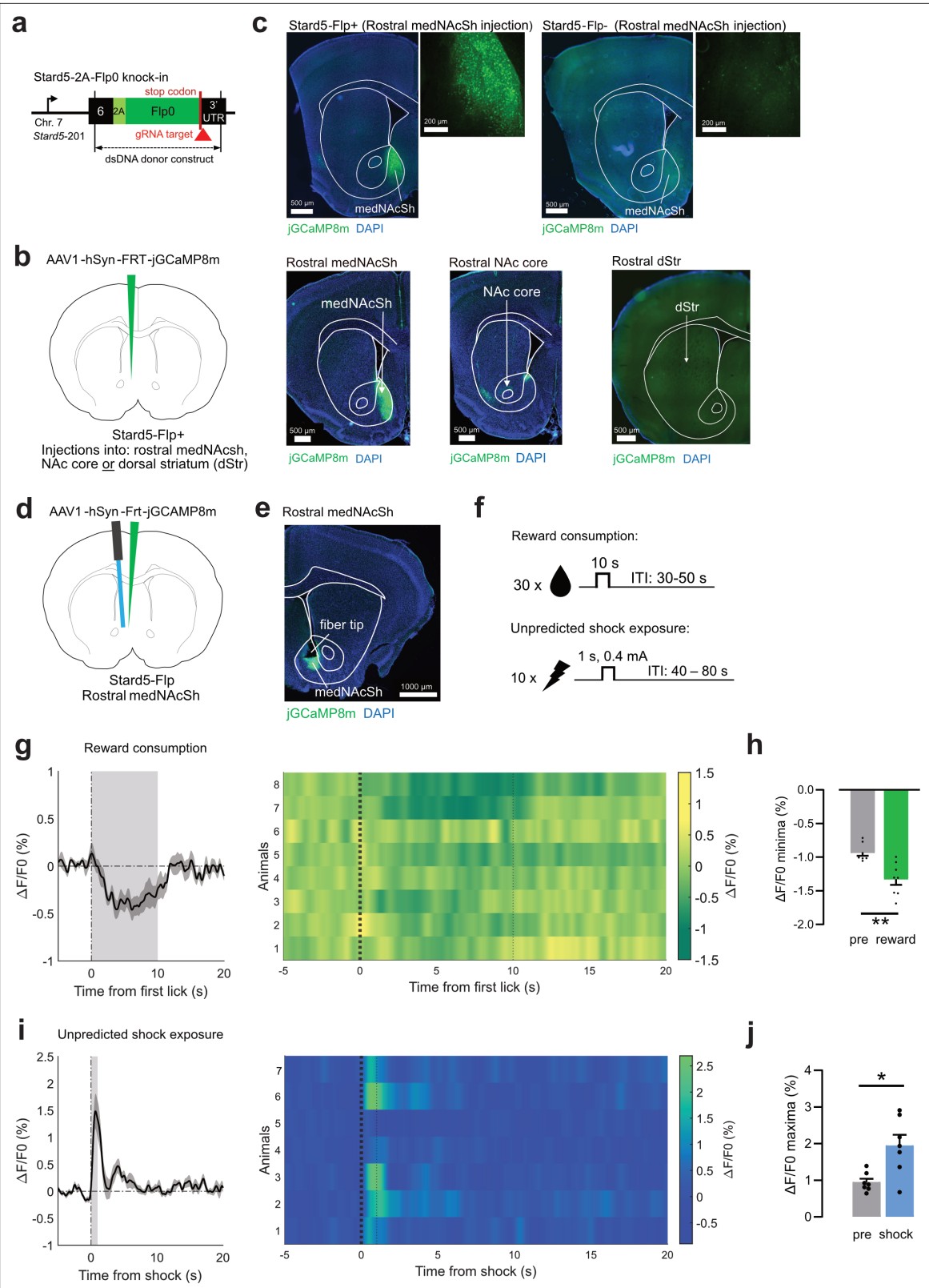

**Figure 7.** Stard5 is a molecular marker for the rostral medNAcSh feeding hotspot. (**a**) CRISPR/Cas9-based strategy to generate a Stard5-2A-Flp0 knock-in mouse (abbreviated Stard5-Flp). A double-stranded (ds) DNA knock-in template expressing the Flp0 recombinase (dark green bands), the RAKR and T2A self-cleaving peptide (light green bands), and *Stard5* homology arms spanning exon 6 and the 3'UTR of the *Stard5* gene (black bands) was used to create the model. The guide (g)RNA (red triangle) targets the stop codon (blue) of the *Stard5* gene (Chromosome 7) at its most expressed transcript

*Figure 7 continued on next page*

*Figure 7 continued*

in the NAc (*Stard5*-201; see *Figure 7—figure supplement 1*). Knock-in was achieved by homologous recombination. (**b**) Left: Surgery schematic for the validation of the Stard5-Flp mouse. An AAV expressing a flp-dependent vector (jGCaMP8m) was injected into the rostral or caudal medNAcSh, dorsal striatum, or NAc core of Stard5-Flp mice. (**c**) Top: Representative coronal images and zoom-in showing successful expression of a Flp-dependent fluorophore in the rostral medNAcSh of Stard5-Flp mice, with very minimal expression in Stard5-Flp negative mice (few cells) indicating a minor Flp leak of the AAV. Close to no expression could be detected in the caudal medNAcSh, NAc core, or dorsal striatum (dStr) after injection directly into those regions, confirming that, within the striatal complex, Stard5 expression is primarily confined to the medNAcSh. (**d**) Surgery schematics of Stard5-Flp mice injected with a flp-dependent calcium indicator (FRT-jGCaMP8m) and implanted with unilateral optic fibers in the rostral medNAcSh to image calcium photometry signals. (**e**) Representative coronal image of rostral medNAcSh Stard5-Flp mice expressing FRT-jGCaMP8m and implanted with an optic fiber. (**f**) Top: Protocol for the unpredicted reward behavioral task: 30 rewards were available for 10 s at random intervals. Bottom: Protocol for the unpredicted shock behavioral task: 10 shocks (0.4 mA) were available for 1 s at random intervals. (**g**) Left: Average calcium activity in *Stard5*+ neurons (all trials) shown as normalized fluorescence ($\Delta F/F0$, %) in the rostral medNAcSh over time aligned to the onset of reward consumption, that is first lick onset (0 s). Grey shading represents the ~10 s epoch for reward consumption. Right: Heatmap showing calcium activity across individual trials (each row represents an animal) aligned to reward consumption. Thin dotted line: end of reward access. The average number of successful lick trials was 18.5 out of 30 (62%). (**h**) Quantification of the data in (**g**) depicts a significant decrease in rostral medNAcSh Stard5 cell activity upon reward consumption, as shown by a significant decrease in $\Delta F/F0$ minima in the 0–5 s reward epoch vs. –6 to –1 s pre-reward epoch. Paired t-test, t(7) = 4.630, p=0.0024. N=8 mice, 4 female, 4 male. (**i**) Left: Average calcium activity in *Stard5*+neurons (all trials) shown as normalized fluorescence ($\Delta F/F0$, %) in the rostral medNAcSh over time aligned to shock onset (0 s). Grey shading: shock epoch. Right: Heatmap showing calcium activity across individual trials (each row represents a trial) aligned to shock onset. Thin dotted line: end of reward access. (**j**) Quantification of the data in (**i**) depicts a significant increase in rostral medNAcSh Stard5 cell activity upon shock exposure, as shown by a significant increase in $\Delta F/F0$ maxima in the 0–5 s shock epoch vs. –6 to –1 s pre-shock epoch. Paired t-test, t(6) = 3.305, p=0.0163. N=7 mice, 1 female, 6 male. Data is mean ± SEM. *p<0.05; **p<0.01; ***p<0.001. See also *Figure 7—figure supplements 1–3*.

The online version of this article includes the following figure supplement(s) for figure 7:

**Figure supplement 1.** Stard5-201 is the dominant splice variant expressed in the NAc, making it an adequate genomic region for 2A-Flp0 vector targeting (Stard5-Flp mouse line generation).

**Figure supplement 2.** The Stard5-Flp transgenic mouse line genetic modification does not show off-target in vivo effects.

**Figure supplement 3.** The Stard5-Flp mouse line allows to identify projection sites of Stard5 cells.

to the three classical projection targets of medNAcSh D1-SPNs (see *Figure 1—figure supplement 1*) and D2-SPNs. Stard5-Flp mice were then injected with an AAV expressing a Flp-dependent calcium indicator (jGCaMP8m) and optic fibers implanted above the rostral medNAcSh to record the population activity of Stard5 + cells (*Figure 7d and e*). Similarly to rostral medNAcSh D1-SPNs, we found that Stard5 + cells pause during reward consumption (*Figure 7f–h*), and get acutely excited upon exposure to aversive stimuli (*Figure 7f and i–j*). This confirms that Stard5 + cells recapitulate some of the behavioral phenotypes of rostral medNAcSh D1-SPNs. It further suggests that Stard5 is a molecular marker of the rostral medNAcSh feeding hotspot.

## Discussion

In recent years, the medNAcSh has gained increasing appreciation as a key brain region that controls hedonic feeding (reviewed in *Marinescu and Labouesse, 2024*). At the cellular level, studies have shown that D1-SPNs are one of the primary cellular mediators of these effects as they can rapidly and powerfully regulate the amount of food consumed in rodents independent of metabolic need (*O'Connor et al., 2015*; *Thoeni et al., 2020*; *Bond et al., 2020*; *Maldonado-Irizarry et al., 1995*; *Stratford and Kelley, 1997*). At the topographical level, in situ pharmacological studies have indicated the existence of a rostro-caudal gradient in the medNAcSh, suggesting that its rostral but not caudal subregion is the primary hotspot for feeding regulation (*Castro et al., 2016*; *Castro and Berridge, 2014b*; *Krause et al., 2010*; *Reed et al., 2018*; *Reynolds and Berridge, 2008*; *Reynolds and Berridge, 2002*; *Richard and Berridge, 2011*). However, the cellular, circuit, and molecular mechanisms of this gradient were, this far, unclear. Here, we combined fiber photometry, optogenetic, and neuroanatomical experiments, identifying a strong functional dichotomy between rostral and caudal medNAcSh D1-SPNs in feeding behavior, but not in aversive assays, thus demonstrating the existence of a rostral medNAcSh D1-SPN feeding hotspot. Using open-source ISH, scRNAseq datasets, and in-house FISH experiments, we show that functional gradients in the medNAcSh are paralleled by molecular gradients along the rostro-caudal axis, delineating the existence of *Stard5* and *Peg10* as markers of the rostral and caudal medNAcSh, respectively. Finally, we use our newly developed

Stard5-Flp driver mouse line, finding that Stard5 cells replicate functional phenotypes of the rostral D1 feeding hotspot. This work provides insights into the functional and molecular organization of the medNAcSh in the context of feeding and aversion behavior and delineates a novel marker, *Stard5*, to access the rostral medNAcSh feeding hub. Importantly, our results indicate that spatial organization also defines functional specialization in the medNAcSh, and that molecular markers such as Stard5 provide access to these spatially defined subterritories rather than labeling a single, homogenous neuronal subtype.

Our work extends previous research delineating a behavioral and anatomical dichotomy between rostral and caudal sections of the medNAcSh (*Baumgartner et al., 2020*; *Castro et al., 2016*; *Castro and Berridge, 2014a*; *Faure et al., 2010*; *Krause et al., 2010*; *Pedersen et al., 2022*; *Reed et al., 2018*; *Reynolds and Berridge, 2008*; *Reynolds and Berridge, 2002*; *Richard and Berridge, 2011*) and aligns with similar concepts demonstrated in the dorsal striatum (*Hintiryan et al., 2016*; *Valjent and Gangarossa, 2021*). In our photometry work, we show that at the population level, rostral medNAcSh D1-SPNs pause, while caudal medNAcSh are acutely excited upon reward consumption. This finding corroborates past work showing that forebrain excitatory inputs into the medNAcSh pause during consumption at rostral but not caudal medNAcSh sites (*Reed et al., 2018*) and that individual D1- and D2-SPNs show rostro-caudal gradients in their responses to feeding (*Pedersen et al., 2022*). We also find that optogenetic stimulation of D1-SPNs in the rostral medNAcSh suppresses reward consumption, while there was a trend for an eating-inhibitory effect that was not significant in the caudal medNAcSh, suggesting the existence of a gradient rather than a binary on/off phenomenon. In addition, comparison of licking behavior during the laser-off blocks revealed an interesting effect: following cessation of opto-stimulation, rostral ChrimsonR mice licked more than caudal ChrimsonR and control mice, suggesting a possible compensatory overconsumption. One possible interpretation is that the optogenetic parameters used suppressed consummatory behavior without reducing the motivation to obtain the reward. Furthermore, consistent with the RTPPA results, activation of rostral D1-SPNs may be experienced as aversive and termination of the optogenetic stimulation could produce relief, which in turn reinforces the licking behavior. Further investigations are required to test these possibilities. Importantly, our behavioral effects could not be explained by reduced locomotor activity; indeed, locomotion was either unchanged or increased in a small cohort of rostral ChrimsonR mice following optogenetic stimulation. The increased locomotion likely reflected appetitive behavior and is consistent with past chemogenetic studies (*Zhu et al., 2016*).

Of note, in this paper we decided to use the term 'rostro-caudal gradient', motivated by converging evidence from prior pharmacological studies (see below) and scRNA sequencing data (*Chen et al., 2021*; *Stanley et al., 2020*), which show continuous molecular and functional changes along the rostro-caudal axis of the medNAcSh rather than sharply defined boundaries. Our use of the term 'gradient' therefore reflects this established molecular organization, even though our functional experiments sampled only two representative positions along this continuum. Our findings align with previous studies identifying rostro-caudal gradients in the medNAcSh, which found that in situ infusion of the glutamate antagonist DNQX or the GABA receptor agonist muscimol in the rostral medNAcSh promotes appetitive responses like feeding. The DNQX effects were also suppressed by a D1 but not D2 antagonist (*Reynolds and Berridge, 2001*; *Richard and Berridge, 2011*).

Interestingly, studies also found that DNQX or muscimol infusions into the caudal, but not rostral, medNAcSh promote aversive behaviors indicating the existence of rostro-caudal gradients also for aversion (*Reynolds and Berridge, 2001*; *Richard and Berridge, 2011*). Here, we further tested the existence of rostro-caudal gradients for aversion, asking whether D1-SPNs in the rostral vs. caudal medNAcSh respond differently to aversive stimuli. To ensure that any observed effects were not specific to a single stressor, we tested two distinct aversive stimuli (foot shock and tail lift). In both cases, we found no rostro-caudal differences, as D1-SPNs in both subregions responded with excitation. We also asked whether optogenetic activation of these neurons is perceived as aversive. Stimulation of D1-SPNs in both rostral and caudal medNAcSh promoted aversive behavioral responses in the RTPPA experiment. Hence, in contrast to the pharmacological inhibitions mentioned above, we did not detect differences in aversive behaviors according to the rostro-caudal medNAcSh site. Here again, our behavioral effects were not explained by reduced locomotor activity as no locomotion differences were detected in the RTTPA.

The lack of divergent responses within subregions of the medNAcSh in aversion contrasts with published work identifying differences in either the photometry activity patterns or the RTTPA functional responses when comparing medial to lateral NAcSh (*Chen et al., 2023*) or medNAcSh D1-SPN projections to VP vs. VTA (*Liu et al., 2022*). In our study, the lack of subregional differences could be due to the fact that our injection sites in rostral and caudal medNAcSh were too close to each other to distinguish region-specific effects. However, in a subset of animals, we validated and confirmed accurate injection and optic fiber placements within the intended medNAcSh subregions. Moreover, since we did detect significant differences between rostral and caudal medNAcSh in the reward consumption photometry and optogenetic assays, this suggests that injection site proximity was not a limiting factor. Future studies using minimal AAV injection volumes and/or tapered optic fibers (*Pisanello et al., 2017*) would help to more precisely restrict the manipulation, thereby enabling refined dissection of spatially confined subregions. Another interesting possibility is that D2-SPNs mediate such rostro-caudal aversive gradient, a cell type we did not manipulate in the present study, which represents a limitation of this work. This possibility would align with previous work showing that the modulation of aversive behaviors by the caudal medNAcSh requires both D1Rs and D2Rs (*Richard and Berridge, 2011*).

Intriguingly, past studies found that rostro-caudal gradients are plastic, that is they can be retuned by the valence of the environment. While in a standard test chamber, rostral DNQX promotes feeding and caudal DNQX promotes aversion, the feeding zone expands caudally in a calm, home-like condition, while the aversion zone expands rostrally in a highly stressful condition (*Baumgartner et al., 2020*; *Reynolds and Berridge, 2008*; *Richard et al., 2013*; *Richard and Berridge, 2011*). Future work could address whether the same environmental retuning impacts the behavioral effects of medNAcSh D1-SPNs, and in turn what are the possible molecular or circuit mediators of these effects. Moreover, it would be interesting to determine if the food consumption effects of D1-SPN optogenetic manipulation are related to the avoidance effects in the RTPPA assay, that is is the inhibition of food consumption due to an internal state of aversion. Since we show that optogenetic stimulation of the caudal medNAcSh induces aversion without suppressing food consumption, this suggests that aversive signaling and feeding behavior can be dissociated. Instead, our findings support the idea that D1-SPNs in the rostral medNAcSh function as a neural hub at the intersection of reward processing and motor (consumption) output. Future studies should dissect this further.

We here determined the projection patterns of medNAcSh subregions and find that the rostral and caudal medNAcSh both project to the LH (*Figure 1—figure supplement 1*), an important feeding center (*Rossi, 2023*). Although in this study we did not quantify the density of projections, previous data identified a stronger medNAcSh to LH projection arising from rostral as compared to caudal medNAcSh sections (*Thoeni et al., 2020*). Such differential projections to the LH could explain behavioral differences between these two brain subregions as multiple studies have demonstrated a dominant role for the medNAcSh to LH projection in the D1-SPN eating effects (*O'Connor et al., 2015*; *Stratford and Kelley, 1999*; *Thoeni et al., 2020*). Since both LH GABA and LH glutamate neurons receive D1-SPN input (*Liu et al., 2024*; *O'Connor et al., 2015*; *Thoeni et al., 2020*; *Zhou et al., 2022*), future studies could investigate the relative contribution of these two cell types in the feeding effects of rostral medNAcSh D1-SPNs. The contribution of rostral medNAcSh D1-SPN projections to the VP and VTA (*Bond et al., 2020*; *Pedersen et al., 2022*) should also be addressed in future work. Moreover, whether D2-SPNs also regulate behavior differentially according to a rostro-caudal axis remains to be determined. Optogenetic activation of D2-SPNs in the medNAcSh did not affect feeding (*O'Connor et al., 2015*), but it is possible that rostro-caudal differences emerge when restricting manipulations to either subregion. Moreover, a recent preprint indicated that medNAcSh D2-SPNs, in particular its projections to the VP, regulate reward preference (*Pedersen et al., 2022*), while other studies have shown that the medNAcSh regulates not only food consumption but also food liking, food wanting (*Castro and Berridge, 2014b*), and extinction in a Pavlovian task (*Domingues et al., 2025*). Additionally, a new study showed that manipulation of D2-SPN cell bodies in the medNAcSh modulates reward preference, self-stimulation, and palatable food intake in a frequency- and context-dependent manner (*Requejo-Mendoza et al., 2025*). Together, these findings suggest that D1- and D2-SPNs within the medNAcSh play complementary rather than opposing roles in reward processing. Hence, the potential role of rostral and caudal medNAcSh D1- and D2-SPNs in food-related behaviors beyond the act of consumption could be addressed in future work.

Other possible circuit mechanisms could explain the rostro-caudal functional differences identified here, in particular the differential circuit inputs into the rostral vs. caudal medNAcSh. It is well established that the medNAcSh receives strong excitatory input from the ventral hippocampus (vHipp), amygdala (Amy), paraventricular nucleus of the thalamus (PVT), and prefrontal cortex (PFC; reviewed in *Marinescu and Labouesse, 2024*). Interestingly, previous work found that these forebrain regions innervate the rostral and caudal medNAcSh via distinct cellular subpopulations (*Reed et al., 2018*). vHipp inputs to the rostral medNAcSh were also found to be stronger as compared to the caudal medNAcSh (*Trouche et al., 2019*). Moreover, optogenetic inhibition of excitatory inputs into the rostral medNAcSh promotes feeding, while similar phenotypes are not detected in the caudal medNAcSh (*Reed et al., 2018*). Hence, it is very likely that differential innervation patterns into the rostral vs. caudal medNAcSh explain the emergence of a rostral feeding hotspot. Previous work demonstrated that, in addition to excitatory forebrain inputs, the VP also sends strong inhibitory inputs into the medNAcSh and directly regulates feeding (*Vachez et al., 2021*); future work could address the rostro-caudal innervation patterns of these VP arkypallidal cells and their contributions to functional differences.

Having established a robust functional dichotomy of D1-SPNs along the rostro-caudal axis in reward consumption, we next asked whether this functional organization is mirrored by differences in molecular composition across the medNAcSh. Using multiple anatomical techniques, we find strong differences in the molecular composition of the rostral vs. caudal medNAcSh, which in turn could explain behavioral differences between these brain subregions. Indeed, we detected multiple markers showing enriched expression in the rostral vs. caudal medNAcSh. In particular, we identified and later validated *Stard5* as a molecular marker labeling the rostral medNAcSh and Peg10 the caudal medNAcSh, respectively. Future work using spatial transcriptomics could extend these findings to identify further relevant rostro-caudal markers at the transcriptome-wide level (*Piwecka et al., 2023*). Importantly, we find that molecular markers do not exhibit a binary on/off expression pattern but rather demonstrate a gradient-like pattern, being enriched vs. depleted in specific subregions of the NAc. This aligns with previous scRNAseq and MERFISH studies, highlighting the existence of spatial gradients of molecularly defined cells within the NAc or dorsal striatum (*Chen et al., 2021*; *Stanley et al., 2020*; *Zhang et al., 2023a*). Authors outlined the existence of two types of markers: (1) cluster markers that selectively mark rare and unique cellular subtypes and (2) spatial markers that are broadly expressed in a spatially restricted manner within NAc subregions. One example of the former are Serpinb2 and Tac2, which labeled small and separate clusters of D1-SPNs in the medial NAc (*Chen et al., 2021*) and regulate behavior in a non-canonical manner (*Liu et al., 2024*; *Zhao et al., 2022*). Here, we build on past findings (*Chen et al., 2021*; *Stanley et al., 2020*), showing that *Stard5* is one of the latter, namely a spatial marker for a spatially defined subterritory, the dorsal part of the rostral medNAcSh. Rather than labeling a unique rare cell type, *Stard5* expression is almost ubiquitous within D1- and D2-SPNs in that subregion (up to 80% of SPNs), making it a noteworthy marker to broadly target the projections of this functionally unique NAc subregion, the rostral medNAcSh feeding hotspot. This makes Stard5 a spatial molecular landmark that captures the cellular ensemble of the rostral feeding hotspot, rather than a marker defining a single functional cell class. It is interesting that Stard5, a START-domain protein implicated in cholesterol metabolism and cellular stress responses (*Alpy and Tomasetto, 2005*; *Rodriguez-Agudo et al., 2012*; *Calderon-Dominguez et al., 2014*), and *Peg10*, an imprinted gene with roles in embryonic development and cancer (*Mou et al., 2025*), mark distinct rostro-caudal domains of the medNAcSh. Whether these genes themselves causally contribute to appetitive and consummatory behaviors, or aversive processing in this region remains an important question for future studies. Using our new Stard5-Flp mouse driver line, we confirm that rostral medNAcSh Stard5 cells recapitulate feeding-related phenotypes of rostral but not caudal medNAcSh D1-SPNs, confirming that molecular markers and spatial location intersect to define cellular function. One important caveat of the Stard5-Flp mouse is that it targets not only D1- but also D2-SPNs, as well as other cell types (primarily interneurons) in the rostral medNAcSh. This heterogeneity does not undermine the utility of the Stard5-Flp line as a targeting strategy, but it reflects the broader principle that the functional specialization arises also from spatially organized ensembles rather than strictly cell-type-restricted populations. However, the Stard5-Flp line was designed using a Flp recombinase so that it could be crossed with selected cre lines to label more restricted cell types. This strategy could be used in future work to dissect the contribution of D1+vs. non-D1+Stard5 cells

in the phenotypes we describe herein, something we could not address in the current study, which represents a limitation.

Together with multiple other studies (*Chen et al., 2023*; *Liu et al., 2024*; *Zhao et al., 2022*), our findings suggest that the molecular composition of NAc cellular clusters or NAc subregions strongly dictate their function. This aligns with recent work showing that other cellular properties such as spatial location, circuit inputs, or circuit outputs also regulate function (*Al-Hasani et al., 2015*; *Chen et al., 2023*; *Liu et al., 2024*; *Zhou et al., 2022*). For instance, recent work finds that medNAcSh cells receiving PVT input drive aversion via LH GABA neurons, while those receiving BLA input drive reinforcement via VTA GABA and LH glutamate cells (*Zhou et al., 2022*). Thus, the NAcSh is composed of intermingled neural circuits that can dictate behavioral outcomes via multiple and sometimes opposing sub-circuits. The finding that NAc molecular composition strongly regulates function represents an important piece of information with critical implications for future therapeutic strategies. Indeed, if druggable targets such as receptors, transporters, or enzymes can be identified as molecular markers labeling unique NAc subregions or unique NAc cell clusters, this could allow developing targeted therapies to increase or decrease reward consumption in a specific manner by up- or downregulating the activity of unique NAc cells.

Our study has potential applications in applied and therapeutic research. First, it could inform future studies investigating the neuronal regulation of body weight and energy balance by the NAc. In this study, we show that the rostral medNAcSh acutely regulates food intake: it would be interesting to determine if these effects are persistent over several days and, in turn, if chronic modulation is sufficient to trigger changes in body weight or body composition. Although these experiments have not been conducted, previous work showed that synaptic plasticity at medNAcSh D1-SPN terminals in the LH regulates overeating over more than 24 hr (*Thoeni et al., 2020*). Moreover, we and others have used broader manipulations of the NAc or dorsal striatum, showing that these brain regions as a whole can powerfully impact body weight, energy metabolism, and obesity risk (*Labouesse et al., 2018*; *Matikainen-Ankney et al., 2023*; *Walle et al., 2024*). Hence, future work could determine whether the rostral medNAcSh plays particularly important roles in these effects. In addition, the NAc is implicated not only in food intake regulation but also in neuropsychiatric diseases like depression or addiction (*Lüscher and Janak, 2021*). Dissecting which subregions of the NAc regulate which types of behavior could have important implications for the understanding of the pathophysiology of these brain disorders. For instance, it would be interesting to determine whether the addiction-related impact of NAcSh D1-SPNs or incoming inputs also follows a rostro-caudal gradient (*Corre et al., 2018*; *Gibson et al., 2018*). The NAc is the target of multiple therapeutic agents, including anti-depressants (*Xu et al., 2020*) or anti-obesity drugs, including glucagon-like peptide 1 (GLP-1) mimetics (*Adan, 2013*; *Müller et al., 2022*). Better delineating the cellular targets of these compounds could support the further refinement of these therapeutic agents into drugs with lesser side effects or stronger impact, for example by inhibiting hedonic eating to support weight loss or weight maintenance in the context of obesity. Although there are GLP-1 receptors in the NAc (shell and core), peripheral GLP-1 mimetics do not reach these brain regions due to limited blood-brain-barrier permeability (reviewed in *Marinescu and Labouesse, 2024*). Since the medNAcSh can get activated by GLP-1 agonists (see e.g. *Labouesse et al., 2012*), these effects are likely occurring via indirect circuit effects. Moreover, the NAc is an important candidate for deep brain stimulation (DBS) treatment, in particular for obsessive-compulsive disorder or depression (*Bewernick et al., 2010*; *Creed et al., 2015*; *Nho et al., 2024*), but also disorders of dysregulated eating (*Hounchonou et al., 2023*; *Shivacharan et al., 2022*; *Zhang et al., 2015*). Describing a topographical map of how NAc subregions control various types of behavior could support the refinement of electrode targeting strategies in the context of DBS-based therapy.

In summary, our study provides clues into the cellular, circuit, and molecular mediators of a rostro-caudal feeding gradient, identifying a D1 rostral hotspot in the medNAcSh with critical roles in feeding behavior. We describe a molecular marker, *Stard5*, that allows labeling the rostral medNAcSh feeding hotspot using a new Flp driver line. Our work also provides insights into the functional organization of the medNAcSh, demonstrating the importance of functional and molecular gradients. This could inform more localized targeting of future therapeutic interventions aiming at treating disorders of dysregulated eating.

## Methods

### Mice

Adult (>8 weeks old) male and female C57Bl6/J mice, Drd1-cre (B6.FVB(Cg)-Tg(Drd1-cre)EY262Gsat/Mmucd) and Stard5-2A-Flp0 (C57BL/6J-Stard5(2A-FLPo)Em1Ethz/Mlab) (Stard5-Flp, generated for this paper; see below) backcrossed onto C57BL/6 J background were used for experiments. All animals were housed at 22–25°C under a reversed 12 hr light-dark cycle (dark 8 am to 8 pm) fed with a regular chow diet (18.5% proteins, 4.5% fat, 4.5% fibers, 54.2% digestible carbohydrates; 3.85 kcal/g gross energy, #3437, Kliba-Nafag, Switzerland). Behavioral testing was done in the dark phase unless otherwise indicated. Animals for behavior were handled and, if necessary, habituated to patch cord tethering before testing. All animal procedures were performed in accordance with the Animal Welfare Ordinance (TSchV 455.1) of the Swiss Federal Food Safety and Veterinary Office and were approved by the Zurich Cantonal Veterinary Office or other relevant National and Institutional regulatory bodies.

### Generation of the Stard5-2A-Flp-mice via Crispr/Cas9 genetic engineering

Analysis of Stard5 transcript expression in the striatum

We set out to design a Stard5-2A-Flp0 mouse line by inserting a 2A-Flp0 sequence to the 3'UTR of the *Stard5* gene (prior to the stop codon). The *Stard5* gene has two differential splicing variants that contain open-reading frames (protein coding): Stard5-201 and Stard-202, which have different 3'UTR regions and terminal exons. To ensure 2A-Flp0 is targeted to a variant that is expressed in the NAc, we analyzed a bulk RNAseq dataset (n=2 samples) of actively translating transcripts in Stard5-expressing cells of the NAc generated using a Stard5-TRAP mouse line (BioProject PRJNA575143 *Heintz, 2019*). Paired-end FASTQ files were downloaded from the SRA using the `fastq-dump` function of the sratoolkit. Read quality was inspected using FastQC (*Andrews, 2026*). Quality-based and length-based filtering and adapter removal were performed using fastp (*Chen et al., 2018*). Alignment was performed using the STAR aligner (*Dobin et al., 2013*) using the GENCODE (*Frankish et al., 2021*) mouse genome build GRCm39 (release M28). The output BAM files were filtered to the Stard5 region, and track plots generated in R using the package Gviz (*Hahne and Ivanek, 2016*). Sashimi plots were generated to visualize the total counts of transcripts that align to specific exons (histograms) and predicted splice junctions (curved lines). The sequencing library preparation was unstranded, preventing us from determining whether reads come from the sense or antisense strand. Hence, some of the Stard5 reads on the sense strand in the last exon of the *Stard5*-202 variant overlap with the IL16 gene on the antisense strand, making it impossible to deconvolute reads that map to Stard5 or IL16 in those overlapping regions, leading to artificial splice junctions across *Stard5* and *IL16*. Still, despite these limitations, the results were unambiguous, showing a large proportion of the reads (total splice junction counts) mapping to the last exon of the Stard5-201 isoform (5742 reads mapped across both samples) rather than to the last two exons of the Stard5-202 isoform (246 reads; *Figure 7—figure supplement 1*). Hence, the *Stard5*-201 transcript and its 3'UTR region are an adequate genomic region to be targeted. *Generation of the Stard5-2A-Flp0 transgenic mouse line.* The Stard5-2A-FlpO transgenic mouse line (official strain name: C57BL/6J-Stard5(2A-FLPo)Em1Ethz/Mlab) was created at the Transgenic Core of the ETH Phenomics Center. In this model, the FlpO gene, a mouse codon-optimized version of the FLP gene, was inserted at the C-terminus of the *Stard5* gene. SV40 nuclear localization signal was included at its N-terminus. Gene expression was controlled via a 2 A self-cleaving peptide (T2A) sequence, preceded by a Furin cleavage site (RA^KA), and terminated by the *Stard5* polyA signal. To generate this transgenic line, gRNAs were designed using the CRISPOR algorithm (http://crispor.gi.ucsc.edu/) to target the region near the stop codon of the mouse *Stard5* gene. The selected gRNA sequence was CAATAGGCGACTGTCAGTGA. A targeting template, 1962 bp in length, was used to facilitate homologous recombination (detailed sequence in Annex). This template was dialyzed against OmniPur Water (Merck Millipore) and IDTE buffer (Integrated DNA Technologies) using a hydrophilic MCE filter membrane (0.22 µm pore size, Millipore Merck). The *Stard5* sequence modification was achieved using CRISPR/Cas9 in two-cell stage mouse embryos. Female C57BL/6 J donors were primed for ovulation through intraperitoneal injections of 5 IU Pregnant Mare Serum Gonadotropin (PMSG, Novatein Bioscience),

followed 48 hr later by 5 IU Human Chorionic Gonadotrophin (hCG; Chorulon, MSD Animal Health). Following hCG administration, females were mated with males of the same strain, and zygotes were collected from oviducts. Remaining cumulus cells were removed using a 5-min incubation in 0.3% Hyaluronidase (Sigma-Aldrich) in M2 medium (Sigma-Aldrich). Embryos were cultured overnight at 37 °C and 5% $CO_2$ in Embryomax KSOM medium (Sigma-Aldrich). Microinjections were performed on a microinjection system that included an inverted microscope with Nomarski optics (Leica), micromanipulators (Eppendorf), and a FemtoJet microinjection unit (Eppendorf). The injection mix contained IDTE buffer pH 7.5, annealed Alt-R CRISPR-Cas9 crRNA and tracrRNA (50 ng/µl), Alt-R S.p.Hifi-Cas9 Nuclease V3 (50 ng/µl), Alt-R HDR Enhancer V2 (1 uM), and Alt-R HDR Donor Block double-stranded template (20 ng/µl) from IDT. Injection mix was filtered using Ultrafree-MC column (Merck Millipore). After injection, surviving embryos were transferred into the oviducts of 8- to 20-week-old 0.5 days post coitus pseudopregnant CD-1 (RjOrl:SWISS) females mated with vasectomized males. Offspring genomic DNA was extracted from toe clip biopsies using the NucleoSpin Tissue kit (Macherey-Nagel) and analyzed for correct knock-in insertion via PCR using Phusion Plus DNA Polymerase (Thermo Fisher Scientific) and Sanger sequencing (Microsynth AG). The following primers were used for PCR: 5' GCACCTAGCTGTCCTACCAG 3' and 5' GATGTCGGTGATGTCGCTCT 3' (842 bp amplicon spanning the 5' homology arm) and 5' TCAGCAACAGCCTGAGCTTC 3' and 5' TACAGCAACCAGCACACAGG 3' (1403 bp amplicon spanning the 3' homology arm). Animals from the F2 generation onwards were used in experiments.

## Metabolic cages and body composition of Stard5-Flp mice

Body composition was assessed using a quantitative nuclear magnetic resonance (NMR) scanner (Echo MRI-100H Body Composition Analyzer, Echo Medical Systems, Houston, USA) as previously described (*Labouesse et al., 2018*). Scans were performed by placing animals into a plastic cylinder, where they were kept immobile (no anesthesia) for the duration of the scan (20 s). Scans computed estimations of fat mass (all fat molecules in the body) and lean mass (all body parts containing water, excluding fat, bone minerals, free water, and hair or cartilage). Indirect calorimetry measurements were performed with the Promethion metabolic cages (Sable Systems International) according to the manufacturer's guidelines and protocols. Energy expenditure analysis was performed by the Promethion software version using the Weir equation (*de V Weir, 1949*): EE = 3.9 x VO2+1.1 x VCO2 (kg/cal) where V02 and VC02 are the volume of oxygen and carbon dioxide (L) consumed and produced, respectively. The respiratory quotient was estimated by calculating the ratio of CO2 production to O2 consumption. Locomotor activity (infrared photobeam based monitoring), food, and water intake were monitored throughout the whole measurement. Recordings were done at 22 °C. Animals were single-caged. Data were collected and averaged from 3 consecutive days after an initial 1–3 days acclimatization period to the cages and the single-caging. Data was analyzed using the CalR application (*Mina et al., 2018*).

## Surgical Procedures

Stereotaxic surgery was performed on a mouse stereotaxic apparatus (Stoelting) using glass pipettes pulled on a Universal Puller (DMZ Zeitz), a Nanoject II (Drummond), and a stereomicroscope (Kaps Optik SOM 82). Fibers were secured in place with miniscrews (FST), iBond (Demadent), dental cement (Charisma A2, Kulzer and Tetric EvoFlow, Ivoclar Vivadent; Demadent) cured with UV light (Woodpecker), and the skin closed with Vetbond (3 M). Photometry fibers (2.5 mm, 400 µm, 0.5 NA; R-FOC-BF400C-50NA) and optogenetic fibers (1.25 mm, 200 µm, 0.22 NA; R-FOC-L200C-22NA) were purchased from RWD. Drd1-cre mice were injected unilaterally with 230 nL of AAV9-CAG-Flex-Synaptophysin-jGCaMP8s for tracing (self-production via ETH VVPP, order 037, titer 2.5x10[12], originally generated in *Labouesse et al., 2023*) or AAV5-hSyn-Flex-jGCaMP8m for photometry (VVF Zurich v628-5, titer 8.1x10E12 vg/mL, diluted 2 x in PBS), at AP +1.5, ML +0.5, DV –4.6, 4.4, 4.2 mm from bregma for the rostral medNAcSh and AP +1.0, ML +0.6, DV –4.8,–4.6, –4.5 for the caudal NAcSh. Photometry fibers were implanted at DV –4.3 (rostral medNAcSh) or DV –4.6 (caudal medNAcSh). For optogenetics, bilateral injections of 300 nL AAV5-hSyn-Flex-ChrimsonR (VVF Zurich v298-5, titer 5.2x10E12 vg/ml, undiluted) or AAV5-hSyn-Flex-mCherry (VVF v116-5, titer 7.5x10E12, undiluted) were done at AP +1.5, ML +0.5, DV –4.6,–4.4, –4.2 (rostral medNAcSh) or AP +1.0, ML +0.5, DV –4.6,–4.4, –4.2 (caudal medNAcSh) on both sides. Optogenetic fibers were implanted at the same

AP, ML location at DV –4.2 [straight] and at AP +1.5, ML 0.5, DV- -4.2 with an angle of 13.4° [angled] (rostral medNAcSh) and DV –4.2 [straight] and AP +1.0, ML 1.5, DV –4.2 with an angle of 13.4°[angled] (caudal medNAcSh). Stard5-Flp were injected with 300 nL of AAV2-CAG-dFRT-ratSynaptophysin-eGFP (VVF 780–2, titer 4.5x10E12 vg/mL, undiluted) and AAV1-hSyn-FRT-jGCaMP (VVF 740–1, titer 6.4x10E12 vg/ml, 2 x diluted in PBS) for tracing the projections (*Figure 7—figure supplement 3*) and for validation experiments/photometry, respectively, at the same coordinates as for the rostral medNAcSh above; or into the NAc core (AP 1.3, ML 1.35, DV –4.4) or dorsal striatum (AP +0.9, ML 1.5, DV –3.5). Behavioral experiments began 4–6 weeks after surgery. Animals for neuroanatomy were prepared >4 weeks after surgery. AAV and fiber locations were validated post-hoc.

## Immunohistochemistry and neuroanatomical tracing

Mice were transcardially perfused with ice-cold 4% paraformaldehyde in PBS under deep anesthesia. For animals with fibers, skulls were immersed into PFA for 2–4 days. Brains were then extracted from PFA-fixed skulls and washed in PBS. Alternatively, brains were extracted after perfusion. Free-floating 50 µm coronal sections were immunostained using the following primary antibodies: chicken GFP (Abcam # ab13970, RRID:AB_300798, used 1:1000), rabbit DsRed (Clontech, #632496, RRID:AB_10013483, used 1:500-1:1000), and secondary antibodies from Thermofisher used 1:1000: 488 goat anti-chicken (A11039, RRID:AB_2534096), 568 goat anti-rabbit (A-11036, RRID:AB_10563566). Digital images were acquired using a Nikon Eclipse T2 epifluorescence microscope or an Olympus/Evident Slideview VS200 Slide Scanner, then processed with ImageJ.

## Analysis of Allen Brain Atlas in-situ hybridization data

We used the open access resource Allen Brain Atlas (https://mouse.brain-map.org/; *Lein et al., 2007*) to identify molecular markers enriched in either the rostral or caudal medNAcSh. Available datasets were filtered for flagged expression in the NAc using the Fine Structure Search 'nucleus accumbens'. The markers included: *4932418E24Rik, Atp6v1c2, Calb1, Carhsp1, Cartpt, Ddit4l, Dlk1, Fxyd7, Gabrg3, Gpr12, Hpcal4, Hs6st2, Ids, Itgb1bp1, Kcng3, Kcnq5, Kctd4, Kctd8, Krt9, Lin7a, Lmo3, LOC381076* [now termed *Baiap3*], *Lrrtm1, Lypd1, Man1a, Mgll, Nrsn1, Pbx3, Peg10, Prkch, Prkg1, Rab27b, Rab3c, Rasal1, Rgs10, Sdc2, Sgsm1, Slc8a1, Smug1, Sorcs2, Stard5, Stra6, Sv2b, Tac2, Th, Zcchc12, Zdbf2, Zdhhc1*. All 48 markers available were screened manually for a visibly strongly enriched expression in the rostral or caudal medNAcSh, also with good expression differences between the medNAcSh and neighboring regions (e.g. NAc core, NAc lateral shell and lateral septum). Quantification of marker expression in the medNAcSh was performed using optical density (OD) calculations in ImageJ. High-resolution coronal images (7–14 images spanning the rostral-caudal axis from AP +1.95 mm to +0.65 mm from bregma) from the Allen Brain Atlas (*Lein et al., 2007*) for 1 dataset (1 animal) were downloaded and OD assessed within the regions of interest (ROI, medNAcSh, NAc core, lateral NAcSh, lateral septum) using 100x100 pixel squares on the raw unedited images. The data from both hemispheres (two measures) were averaged for each ROI. OD values from the ROIs were subtracted from the average OD of the background (taken in the anterior commissure). For the purpose of the figure, images were edited in Microsoft PowerPoint, using contrast and brightness filters applied the same way to all images of the same staining, so as to improve staining visibility.

## Analysis of scRNAseq data

To determine expression patterns of *Stard5* and *Peg10* in the NAc, we utilized a published scRNAseq dataset from the NAc (*Chen et al., 2021*). Gene expression matrix and metadata were downloaded from GEO (GSE118020) and imported into Seurat version 5.1.0 (*Butler et al., 2018*), using the 'min. cells' and 'min.features' parameters set to 3 and 1,250 respectively. Then the following functions were used: 'NormalizeData()' using the 'LogNormalize' method with a scale factor of 1e6; 'ScaleData()' using 'nUMI', 'percent.mito', and 'Sample' as regression variables; 'FindVariableFeatures()' using the 'mean.var.plot' option, and mean and dispersion cutoffs from 1 to infinite. Then, PCA, UMAP, and tSNEs were computed, with UMAP and tSNE using the PCs 1:15. Separate objects for Drd1a+ (D1-SPN) and Drd2+ (D2-SPN) cells were generated, with the 'FindVariableFeatures()' and dimension reductions recomputed for each. Due to their relatively low expression in the dataset, cells were labeled in the Seurat metadata as either Stard5+, Drd1a+, Drd2+, or Peg10+, simply by the presence of a single copy of the respective gene in the main data assay. Feature plots of gene expression were

generated using tSNE and the main data assay, with the 'FeaturePlot()' Seurat function. Proportions of the cells that were labeled as both Stard5 (or Peg10), and either Drd1a or Drd2 positive, were calculated by taking the number of cells that expressed both genes against the total number of cells. Proportions of the Stard5+ (or Peg10+) cells that are also either Drd1a+or Drd2+were calculated by taking the number of Stard5+ (or Peg10+) cells divided by the number of those Stard5+ (or Peg10+) cells that were non-zero for the other gene. Data is pooled from 11 mice, 36,670 cells.

## Fluorescent in situ hybridization assay

*Stard5* and *Peg10* expression patterns in NAc D1- and D2-SPNs were determined by fluorescent multiplex in-situ hybridization (RNAscope, Advanced Cell Diagnostics, CA, USA). Following rapid decapitation of wild-type mice, brains were rapidly frozen in powdered dry ice and stored at –80 °C. Coronal sections of OCT-embedded brains were cut at 20 µm at –20 °C and thaw-mounted onto Super Frost Plus slides (Fisher). Slides were stored at –80 °C until further processing. Fluorescent in situ multiplex hybridization was performed according to the RNAscope Multiplex Fluorescent Reagent Kit v2 and the User Manual for fresh frozen tissue (Advanced Cell Diagnostics, Inc, CA, USA). Briefly, sections were fixed in 4% PFA, dehydrated with increasing percentages of EtOH, incubated with Hydrogen Peroxide, and treated with protease III solution (Advanced Cell Diagnostics, Inc, CA, USA). Sections were then incubated with target probes for mouse *Drd1* to label D1-SPNs (Drd1-C1, Cat No. 406491), mouse *Drd2* to label D2-SPNs (Drd2-C2, Cat No. 406501-C2), mouse *Stard5* to label *Stard5* + cells (Stard5-C3, Cat No. 880931-C3) or *Peg10* to label Peg10 + cells (Peg10-C3, Cat No. 512921-C3). Following probe hybridization, sections underwent a series of probe signal amplification steps followed by incubation of fluorescently labeled probes (TSA fluorophore detection kit, PerkinElmer, MA, USA) designed to target the specified channel associated with *Stard5* or *Peg10* (TSA Vivid Fluorophore 690), *Drd1* (TSA Vivid Fluorophore 520), and *Drd2* (TSA Vivid Fluorophore 570) as a triplex. Slides were counterstained with DAPI, and coverslipped with Fluoromount-G mounting medium (SouthernBiotech, Birmingham, AL, USA). High-resolution images (20 X) were obtained on a fluorescent microscope (Olympus/Evident Slideview VS200 Slide Scanner), exported, and analyzed in Qupath-0.5.1 (**Bankhead et al., 2017**). Annotations were created for both hemispheres for the medNAcSh, lateral NAcSh, and core. Cell detection was performed for each annotation using QuPath's built-in 'Cell Detection' algorithm, with DAPI as the detection channel. The following nucleus parameters were used: background radius 8 µm, median filter radius 0 µm, sigma 1.2 µm, intensity threshold of 1000–3000, and cell expansion of 5 µm. Single measurement classifiers were created for all channels (FITC, Cy3, Cy5) to count the number of cells that were FITC+ (Drd1+only), Cy3+ (Drd2+only), Cy5+ (Stard5+or Peg10+), or double/triple-positive. For each brain, six to eight sections of the medNAcSh were included (covering rostral to caudal bregma levels +1.95 to+0.65 mm; on average two to three sections for most rostral and two to three sections for most caudal sections). Percent labeling was averaged for n=5 mice for Stard5 and n=5 for Peg10.

## Fiber photometry

Fiber photometry was conducted using a RZ10X LUX-I/O Processor with integrated LEDs (Tucker-Davis Technologies) combined with a filter/mirror minicube set (FMC6_IE(400-410)_E1(467–497)_F1(507–554)_E2(566–586)_F2(600–680)_S) (Doric Lenses). The 405 nm LED was passed through a 400–410 nm bandpass filter, the 465 nm LED through a 467–497 nm GFP excitation filter, then both were coupled to a dichroic mirror to split excitation and emission lights. Low-autofluorescence patch cords (400 µm/0.57 NA, Doric) were attached to the optic fibers (400 µm/0.5 NA, RWD) on the mouse's head to collect fluorescence emissions. Signals were filtered through 507–554 nm GFP emission filters coupled to a photodetector on the RZ10X. Signals were sinusoidally modulated, using the TDT Synapse software at 210 and 330 Hz (405 and 465 nm, respectively) via a lock-in amplification detector, then demodulated and low-passed filtered at 6 Hz online. Data were sampled at 1017.3 Hz. 465 nm power at the patch cord was set to 30–35 µW.

## Photometry recordings during reward consumption or aversive stimuli

For behavioral experiments combined with medNAcSh photometry, an iCon behavior control interface (Tucker-Davis Technologies) was used to control an operant box (length 22 cm x width 20 cm x height 38 cm; ENV-307W-CT, Med Associates). The behavioral interface was directly connected

to the RZ10X processor and controlled by custom-written codes written in Pynapse (version 96), a Python-based coding interface integrated into the TDT Synapse software. The iCon consisted of two modules – an iM10 multi-function interface and an iH16 28 V control interface. The iM10 control interface contained the touch detector and received input from the lickometer. The iH16 controlled the retractable lickometer (ENV-352–2 W Retractable Sipper Tube, Med Associates) and the aversive stimulator (ENV-414S, MedAssociates), house light, and house fan. Mouse behavior was monitored using a camera (C270 HD webcam, Logitech) controlled by Synapse. The reward port access was custom-modified using a 3D printer to facilitate reward collection. For unpredicted reward delivery, mice were first trained across 7–10 days. On the first 2 days mice were given unrestricted access to the reward for 10 min followed by 10 min of restricted access with 10 s rewards being accessible at random intertrial intervals (ITIs) of 20–40 s. On days 3–5, mice were trained over 30 trials with the 10 s rewards accessible at random ITIs of 20–40 s. If the animal collected the reward within the specified reward windows, the trial was called a successful trial. If the mouse tried to collect the reward outside of the reward windows, a timeout period of 60 s was introduced during the training sessions. Once the mouse achieved a minimum of 10 successful trials, it was tethered for 2 days and trained similarly to days 3–5. After 10 successful trials, training was repeated for 3 days, with ITIs set at 30–50 s. Upon 12 successful trials, animals were tested in combination with in-vivo fiber photometry recordings under the same conditions. In case of >1 day break just before recordings (e.g. weekends), mice underwent an additional day of training before recording. The learning rate from Day 1 until testing is shown in *Figure 1—figure supplement 2*. For Drd1-cre mice, the experiment was repeated three times (all gave the same results; no learning effects), and the session on the last day was analyzed. For Stard5-Flp mice, the experiment was conducted only once. For unpredicted shock (aversive stimuli #1), mice were exposed (without training) to 10 aversive shocks (0.4 mA, 1 s) with intertrial intervals of 40–80 s. Mice were placed inside a black-and-white striped cylinder in the operant box to create a context distinct from that of the reward experiment. For tail lift (aversive stimuli #2), mice were lifted by the tail 5 times with intertrial intervals of 5 min, in a cage similar to their home cage. The initial lift took an average of 3 s, followed by a period of stabilization in the air for a total of 60 s then mice were returned to the cage floor promptly. The clearest response was found at the acute tail lift; hence that data is shown and analyzed. A 3-min period at the beginning of all sessions with no stimuli was observed, and the data for these 3 min was removed before analysis. MedNAcSh responses to reward consumption or to aversive stimuli were quantified using custom in-house Matlab scripts as done previously (*Labouesse et al., 2024*; *Labouesse et al., 2023*). Events were identified using TTLs within the software. Data was 10 x downsampled to 100 Hz and low-pass filtered (1 Hz). The first 3 min were trimmed to remove fast bleaching artefacts. Change in fluorescence, $\Delta F/F0$ (%), was defined as $(F-F0)/F0 \times 100$, where F represents the fluorescent signal (465 nm) at each time point. F0 was calculated by applying a least-squares linear fit (polyfit) to the 405 nm signal to align with the 465 nm signal, done on fluorescent data obtained in the $-10$ to $+20$ s or 30 s around events. Event-related $\Delta F/F0$ was baseline corrected against the $\Delta F/F0$ average in the $-10$ to $-1$ s time window before event onset. Data was aligned to events of interest, namely the first lick during the 10 s window of reward availability or the shock onset. Average peri-stimulus time histogram (PSTH) curves generated by averaging the data from all mice within a session and heatmaps showing all trials in a session were computed showing the $\Delta F/F0$ values (%) in windows of $-10$ or $-5$ to $+20$ s around events. Two epochs were used for analysis: the baseline epoch spanning $-6$ to $-1$ s before event start and event epoch spanning 0–5 s from event start. Area under the curve (AUC) was calculated for each epoch using the trapezoidal method. Peak maxima and minima $\Delta F/F0$ were obtained by calculating the maxima or minima value for each epoch.

## Optogenetic manipulations in a reward consumption task

Optogenetic stimulation was provided by an amber-light emitting laser (635 nm, Laserglow) and operant boxes and laser were driven using the iCon behavioral interface and custom-written Python code in Pynapse/Synapse software (TDT). Lasers were connected via optical patch cords (200 µm, 0.22 NA) and a rotary joint to the animals' optic fibers. Rewards were available ad libitum via a lickometer (ENV-350CW, Med Associates). Initial training consisted of exposing mice to the lickometer during three to five sessions of 15–25 min. In the first two sessions, mice were not tethered to the patch cord to facilitate training. Once the variability in lick counts across two consecutive sessions was less than 30%, animals were moved to the test session. Two mice did not reach criterion and were excluded

from the experiment. The proof-of-concept experiment consisted of three 8 min blocks, where laser stimulation occurred from minute 8 to minute 16. The stimulation was delivered continuously at 20 Hz (20ms pulses, 10 mW power at fiber tip). The full optogenetic experiment consisted of five 5 min blocks, with laser stimulation during the first, third, and fifth blocks. Optogenetic stimulation was delivered continuously at 20 Hz (20ms pulses, 300 µW power at fiber tip) throughout blocks 1, 3, and 5. All licks were detected with the Synapse software. Data was analyzed using custom-written Matlab codes to generate lick counts and lick counts/min. Additionally, to assess general locomotor activity during the reward consumption task, mice were tracked in the operant box. A small subset of frames (136 frames from 10 video recordings) was manually annotated to define the animal's body center and nose, as well as the four corners of the operant box. These annotations were used to train a YOLO-based pose estimation model (*Redmon et al., 2015*). Locomotion metrics, such as total distance moved, were subsequently derived from the temporal integration of positional data and aligned to opto-on and opto-off epochs of the feeding task.

## Optogenetic manipulations in a real-time place preference and avoidance (RTPPA)

Optical stimulation was provided by an amber-light emitting laser (635 nm, Laserglow), which was activated using Anymaze version 7.37 (Stoelting). Lasers were connected via optical patch cords (200 µm, 0.22 NA) and a rotary joint to the animals' optic fibers. To activate ChrimsonR, we used 20 ms light pulses at 20 Hz with a power of 10 mW (or 300 µW, data not shown, results consistent) at the fiber tip. The RTPPA chamber consisted of two chambers of the same dimension (30 x 30 x 30 cm, width, length, height) with different cues in each chamber. For the main experiment with 10 mW optogenetic laser power (shown in *Figure 4*), one chamber had thick vertical black stripes, while the other had horizontal ones. The experiment was repeated with lower laser power (300 µW) where we found the same results (data not shown): here, one chamber featured a black-and-white checkerboard pattern and the other had diagonal black-and-white lines; experiment performed 5 months after the 10 mW one. Visual cues were also positioned outside the chamber. Testing consisted of three trials of 15 min each on 3 consecutive days. On day 1, animals were exposed to both sides of the chamber, and their preference was determined. Laser stimulation was assigned to the preferred chamber. On days 2 and 3, the laser was stimulated as soon as animals entered the laser-assigned chamber and for the entire duration of their visit. This was triggered in closed loop using a custom-written protocol in the Anymaze software. Time spent and total distance moved in each chamber were computed using Anymaze. Some animals were euthanized early to validate optic fiber surgery implantation; hence not all mice were used in both RTTPA and reward consumption tasks.

## Statistics

Statistical analyses were performed using GraphPad Prism 9 or 10. Data are expressed as mean ± standard error of the mean (SEM). 2-group data were compared with paired or unpaired two-tailed Student's t-tests. Multiple comparisons were evaluated by ANOVA (no Greenhouse correction) and in the case of significant interactions, Sidak post-hoc tests were used. A p-value of <0.05 was considered statistically significant.

## Material availability

The Stard5-Flp mouse is deposited at the ETH Phenomics Center (administration@epic.ethz.ch) in the form of cryopreserved straws (line 3072) under the group of Prof. Ferdinand von Meyenn (ETH). Its CRISPR sequence is available in *Supplementary file 1*.

## Code availability

Original Python code to drive Pynapse is provided in *Supplementary file 2*. Please cite this paper upon usage of the code. No new MATLAB code was developed but the original code can be found in *Labouesse et al., 2024*; *Labouesse et al., 2023*.

## Materials and correspondence

Further information and requests for resources and reagents should be directed to and will be fulfilled by the lead contact, Dr. Marie A. Labouesse (marie.labouesse@hest.ethz.ch).

## Acknowledgements

This project was funded by the Swiss National Science Foundation (SNSF) (Ambizione Grant #PZ00P3_193430 to MAL) with additional support from the Brain and Behavior Research Foundation (Narsad Grant #30854) and the Novartis Foundation for Medical Biological Research (Grant #22B097). MAL received additional funding from the Vontobel Foundation (#1334/2021), the Olga Mayenfisch Foundation and an ETH Zurich Equipment Grant. AMM is funded by a PhD fellowship from the Zurich Neuroscience Center. CLF is funded by the SNSF (Grant #310030_207960). We thank Christian Wolfrum, Johannes Bohacek, and Ferdinand von Meyenn for key infrastructural support; Johannes Bohacek, Theofannis Karayannis, and Christopher Pryce for key scientific input; Ori Bar-Nur for access to fluorescent microscopy; Myrtha Arnold for support in the acquisition of key reagents, Fabienne Rössler and Marcel Schmutz for the locomotion analysis in the reward consumption task. Some images were obtained from SciDraw under a CC-BY license (doi.org/10.5281/zenodo.3925917, doi.org/10.5281/zenodo.5496332, and doi.org/10.5281/zenodo.8319097).

## Additional information

### Funding

| Funder | Grant reference number | Author |
|---|---|---|
| Swiss National Science Foundation | PZ00P3_193430 | Marie A Labouesse |
| Brain and Behavioral Research Foundation | 30854 | Marie A Labouesse |
| Novartis Foundation for Medical Biological Research | 22B097 | Marie A Labouesse |
| Vontobel Foundation | 1334/2021 | Marie A Labouesse |
| Olga Mayenfisch Foundation | | Marie A Labouesse |
| ETH Zurich | | Marie A Labouesse |
| Zurich Neuroscience Center | | Alina-Măriuca Marinescu Marie A Labouesse |
| Swiss National Science Foundation | 310030_207960 | Christelle Le Foll |

The funders had no role in study design, data collection and interpretation, or the decision to submit the work for publication.

### Author contributions

Alina-Măriuca Marinescu, Conceptualization, Data curation, Formal analysis, Validation, Investigation, Visualization, Methodology, Writing – original draft, Writing – review and editing; Eshita Kamal, Validation, Investigation, Methodology; Peter Leary, Data curation, Formal analysis, Investigation, Visualization; Keila Navarro I Batista, Validation; Manuel Klug, Investigation; Nataša Savić, Resources, Validation, Methodology; Christelle Le Foll, Resources, Funding acquisition, Validation; Marie A Labouesse, Conceptualization, Resources, Data curation, Formal analysis, Supervision, Funding acquisition, Validation, Investigation, Visualization, Methodology, Writing – original draft, Project administration, Writing – review and editing

### Author ORCIDs

Alina-Măriuca Marinescu https://orcid.org/0009-0007-4073-0402
Eshita Kamal https://orcid.org/0009-0003-8461-7816
Peter Leary https://orcid.org/0000-0003-1430-6185
Keila Navarro I Batista https://orcid.org/0009-0003-0542-6983
Nataša Savić https://orcid.org/0000-0003-3110-5780
Christelle Le Foll https://orcid.org/0000-0002-6677-5488

Marie A Labouesse https://orcid.org/0000-0002-6850-5852

### Ethics

All animal procedures were performed in accordance to the Animal Welfare Ordinance (TSchV 455.1) of the Swiss Federal Food Safety and Veterinary Office and were approved by the Zurich Cantonal Veterinary Office or other relevant National and Institutional regulatory bodies.

Reviewer #2 (Public review): https://doi.org/10.7554/eLife.108639.3.sa1
Author response https://doi.org/10.7554/eLife.108639.3.sa2

## Additional files

### Supplementary files

Supplementary file 1. Stard5-2A-Flp0 KI template sequence.

Supplementary file 2. Python code in Pynapse for unpredicted reward and unpredicted shock tasks and Optogenetic stimulation during feeding task.

MDAR checklist

Source data 1. Source Data (Separate Excel file).

### Data availability

Source data is provided as *Source data 1*.

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
