## [Editor Report · eLife Assessment]

This study provides a **valuable** contribution to understanding the functional and molecular organization of the medial nucleus accumbens shell in feeding behavior. Through a multimodal approach that integrates in vivo imaging, optogenetic manipulation, and genetic strategies, the authors present **convincing** evidence for rostro-caudal differences in D1-SPN activity, advancing and refining earlier pharmacological frameworks. The discovery of Stard5 and Peg10 as regionally informative markers, together with the introduction of a Stard5-Flp driver line, establishes a foundation for more targeted circuit dissection. While an expanded characterization of other Stard5-positive cell populations (e.g., D2-SPNs, interneurons) would strengthen the work, the experimental rigor and internal consistency of the findings are clear. Overall, this is a technically strong and conceptually meaningful study with broad relevance for those investigating neural mechanisms of reward, affect, and feeding.

---

## [Referee Report · Reviewer #2 (Public review)]

Summary:

Marinescu et al. combine in vivo imaging with circuit-specific optogenetic manipulation to characterize the anatomic heterogeneity of the medial nucleus accumbens shell in the control of food intake. They demonstrate that the inhibitory influence of dopamine D1 receptor-expressing neurons of the medial shell on food intake decreases along a rostro-caudal gradient while both rostral and caudal subpopulations similarly control aversion. They then identify Stard5 and Peg10 as molecular markers of the rostral and caudal subregions, respectively. Through the development of a new mouse line expressing the flippase under the promoter of Stard5, they demonstrate that Stard5-positive neurons recapitulate the activity of D1-positive neurons of the rostral shell in response to food consumption and aversive stimuli.

Strengths:

This study brings important findings for the anatomical and functional characterization of the brain reward system and its implication in physiological and pathological feeding behavior. In the revision, the authors provided additional data that strengthen the specificity of their behavioral effects. It is a well-designed study, technically sound, with clear and reliable effects. The generation of the new Stard5-Flp line will be a valuable tool for further investigations. The paper is very well written, the discussion is very interesting, addresses limitations of the findings and proposes relevant future directions.

Weaknesses:

Identification and characterization of the activity of Stard5-positive neurons will require further characterization as this population encompasses both D1- and D2-positive neurons as well as interneurons. While they display a similar response pattern as D1-neurons, it remains to determine whether their manipulation would result in comparable behavioral outcomes.

---

## [Author Response]

The following is the authors’ response to the original reviews

**Public Reviews:**

**Reviewer #1 (Public review):**
Summary:This study examines how different parts of the brain's reward system regulate eating behavior. The authors focus on the medial shell of the nucleus accumbens, a region known to influence pleasure and motivation. They find that nerve cells in the front (rostral) portion of this region are inhibited during eating, and when artificially activated, they reduce food intake. In contrast, similar cells at the back (caudal) are excited during eating but do not suppress feeding. The team also identifies a molecular marker, Stard5, that selectively labels the rostral hotspot and enables new genetic tools to study it. These findings clarify how specific circuits in the brain control hedonic feeding, providing new entry points to understand and potentially treat conditions such as overeating and obesity.

We thank Reviewer 1 for the positive feedback, summary of our findings and for the thorough reading and constructive comments on the manuscript, which allowed us to improve the quality of the revised version.

Strengths:(1) Conceptual advance: The work convincingly establishes a rostro-caudal gradient within the medNAcSh, clarifying earlier pharmacological studies with modern circuit-level and genetic approaches.(2) Methodological rigor: The combination of fiber photometry, optogenetics, CRISPR-Cas9 genetic engineering, histology, FISH, scRNA-seq, and novel mouse genetics adds robustness, with complementary approaches converging on the central claim.(3) Innovation: The generation of a Stard5-Flp line is a valuable resource that will enable precise interrogation of the rostral hotspot in future studies.(4) Specificity of findings: The dissociation between appetitive and aversive conditions strengthens the interpretation that the observed gradient is restricted to feeding.

We thank Reviewer #1 for their supportive feedback.

Weaknesses and points for clarification(1) Role of D2-SPNs: Since D1 and D2 pathways often show opposing roles in feeding, testing, or discussing D2-SPN contributions would provide an important control and context. Since the claim is that Stard5 is expressed in both D1- and D2MSNs, it seems to contradict the exclusive role of D1R MSNs in authorizing food intake.

We agree that D2-SPNs represent an important and relevant cell population in the context of our study. The Stard5-Flp line labels a mixed population of D1- and D2-SPNs, and we agree that dissecting the distinct contributions of Stard5^+^ D1-SPNs and Stard5⁺ D2-SPNs to feeding behavior would be both interesting and informative.

Although we understand the point raised by the Reviewer, we do not entirely agree that the expression of Stard5 in both D1- and D2-SPNs contradicts the established role of D1-SPNs in authorizing food intake. In the medNAcSh, D1- and D2-SPNs do not exert opposing functions. D2-SPNs project densely to the ventral pallidum and more sparsely to the lateral hypothalamus and, like D1-SPNs, are predominantly rewardinhibited at the population level (Domingues et al. 2025; Pedersen et al. 2022).

We added the following in the discussion: “Additionally, a new study showed that manipulation of D2-SPN cell bodies in the medNAcSh modulates reward preference, self-stimulation, and palatable food intake in a frequency- and context-dependent manner (Requejo-Mendoza et al., 2025). Together, these findings suggest that D1- and D2-SPNs within the medNAcSh play complementary rather than opposing roles in reward processing. Hence, the potential role of rostral and caudal medNAcSh D1- and D2-SPNs in foodrelated behaviors beyond the act of consumption could be addressed in future work.” We also acknowledge that not investigating rostro-caudal gradients of D2-SPN in reward and aversion processing “represents a limitation of this work”.

We fully agree that disentangling the specific contributions of Stard5^+^ D1- and Stard5^+^ D2-SPNs is an important next step. We have now crossed the Stard5-Flp line with Drd1-Cre and A2a-Cre lines. In a pilot experiment (not shown), we injected Flp+,Cre+, Flp+,Cre- and Flp-,Cre+ mice with 4 different FlpOn-CreOn AAVs to determine if any of these AAVs demonstrate specific expression. However, all AAVs exhibited moderate to strong leaky expression of the Cre, preventing reliable cell-type-specific targeting. This was not seen with Flp-only or Cre-only AAVs. The leakiness mentioned is a known challenge of FlpOn-CreOn AAVs and requires additional troubleshooting (e.g. reduce the titer). As this proved to be more challenging than anticipated, this work is ongoing and will be addressed in a future study rather than in the present revisions.

(2) Behavioral analyses:(a) In Figure 2, group differences in consumption appear uneven; additional analyses (e.g., lick counts across blocks and session totals) would strengthen interpretation.

The group differences in consumption that appear uneven likely reflect an overall lower total lick counts per session in the Control group. We have now added analyses on average lick counts per block and session totals in the newly included Supplementary Figure S7, which support the results shown in Figure 2.

Although we observe a difference in total lick count across the entire session between Control and Rostral ChrimsonR mice (Supplementary Figure S7d), we deem the comparison in total session lick counts not that informative here. Instead, we would argue that the laser-on epoch is the most meaningful comparison. During this period, optogenetic activation had no effect on licking behavior in control mice, showed a nonsignificant trend toward reduced consumption in caudal ChrimsonR mice, and produced a significant reduction in lick counts when rostral medNAcSh D1-SPNs were activated (Figure 2g-i and Supplementary Figure S7c).

We added in the discussion the following explanation:

“In addition, comparison of licking behavior during the laser-off blocks revealed an interesting effect: following cessation of opto-stimulation, Rostral ChrimsonR mice licked more than Caudal ChrimsonR and Control mice, suggesting a possible compensatory overconsumption. One possible interpretation is that the optogenetic parameters used suppressed consummatory behavior without reducing the motivation to obtain the reward. Furthermore, consistent with the RTPPA results, activation of rostral D1-SPNs may be experienced as aversive and termination of the optogenetic stimulation could produce relief, which in turn reinforces the licking behavior. Further investigations are required to test these possibilities.”

(b) The design and contribution of aversive assays to the main conclusions remain somewhat unclear and could be better justified.

We appreciate the Reviewer’s comment regarding the design and contribution of the aversive assays. The rationale for including these experiments was to determine whether the rostro–caudal functional segregation observed for reward-related feeding also applies to aversive processing.

First, using foot shock, we tested whether D1-SPNs in the rostral versus caudal medNAcSh respond differently to an aversive stimulus. In contrast to reward-related responses, both populations responded similarly, exhibiting excitation. Second, to ensure that this effect was not specific to a single stressor, we tested a second aversive stimulus (tail lift) and again observed comparable excitatory responses in rostral and caudal D1-SPNs. Third, we assessed whether optogenetic activation of these neurons is perceived as rewarding or aversive. Using a real-time place preference/aversion assay, we found that optogenetic stimulation of D1-SPNs in both subregions induced place aversion.

Together, these experiments show that while D1-SPNs display region-specific effects on reward-related feeding behavior, their activity responses to aversive stimuli and the avoidance response to optogenetic activation are similar across rostral and caudal medNAcSh. This contrast strengthens our conclusion that the D1-SPN rostro-caudal gradient is specific to appetitive contexts.

We added the following in the discussion:

“Here, we further tested the existence of rostro-caudal gradients for aversion, asking whether D1-SPNs in the rostral vs. caudal medNAcSh respond differently to aversive stimuli. To ensure that any observed effects were not specific to a single stressor, we tested two distinct aversive stimuli (foot shock and tail lift). In both cases, we found no rostro-caudal differences, as D1-SPNs in both subregions responded with excitation. We also asked whether optogenetic activation of these neurons is perceived as aversive. Stimulation of D1- SPNs in both rostral and caudal medNAcSh promoted aversive behavioral responses in the RTPPA experiment. Hence, in contrast to the pharmacological inhibitions mentioned above, we did not detect differences in aversive behaviors according to the rostro-caudal medNAcSh site.”

(c) The scope of behavior is mainly limited to consumption; testing related domains (motivation, reward valuation, and extinction) could broaden the significance.

We thank the Reviewer for the suggestion to examine additional behavioral domains such as motivation, reward valuation, and extinction. We focused our efforts on consumption given the large body of literature demonstrating a very important role of the medNAcSh in reward consumption. However, we fully agree that feeding encompasses multiple phases, from appetitive and goal-directed behaviors to consummatory behavior, and that the NAc in general, and to some extent the NAcSh is involved in behaviors across this spectrum. For instance, prior work has shown that the medNAcSh is involved in reward preference and that this follows a rostro-caudal gradient (e.g. Pedersen et al. 2022).

While it would be informative to directly test motivational processes using operant paradigms (e.g., nosepoke or lever-press tasks), our current experimental setup did not allow for these assays. Instead, we performed exploratory experiments manipulating the animals’ internal state with food deprivation. As expected, under food deprivation, total licking increased robustly in control mCherry and Rostral ChrimsonR medNAcSh mice as compared to ad libitum feeding (25 min session with 5 alternating on-off blocks: ad libitum Control = 692 and Rostral ChrimsonR = 1280 average total licks per session, see Figure 2g-h and Supplementary Figure S7d; food deprived Control = 2428 and Rostral ChrimsonR = 2390 total licks averaged for N=9 Control, N = 12 Rostral). Moreover, similar to ad libitum feeding, optogenetic activation of rostral D1-SPNs suppressed licking in food-deprived mice , albeit to a lesser extent than under ad libitum feeding conditions (Figure 2).

These preliminary observations suggest that internal state modulates the role of rostral D1-SPNs in reward consumption, potentially reflecting an interaction between homeostatic and hedonic feeding circuits. However, as this line of investigation was exploratory and not pursued further in the present study, these data are not included in the main manuscript.

**Author response image 1. sa2fig1:** In vivo optogenetic stimulation of rostral medNAcSh inhibits reward consumption to a lesser extent after overnight food deprivation. a. Quantification of the average lick count per 5 min block in mCherry control mice vs. ChrimsonR (rostral) mice, showing a lower lick count in rostral medNAcSh ChrimsonR mice during the opto-stimulation epoch. Blocks of 5 min with or without opto-stimulation were alternated (on/off/on/off/on) for a total of 5 blocks. b. Quantification of mean lick counts in the opto-stimulation vs. non-opto-stimulation epochs shows a significant decrease in lick counts following stimulation of rostral medNAcSh D1-SPNs and no significant difference in the control mice. 2-way RM-ANOVA (group x epoch). Main effects: epoch F (1, 28) = 6.027, p=0.0206; group F (2, 28) = 1.448, p=0.2520; group x epoch F (2, 28) = 8.123, p=0.0017. Sidak post-hoc opto-stimulation vs. non opto-stimulation: Control on vs. off t(28) = 1.856, p=0.2061; Rostral medNAcSh on vs. off t(28) = 3.054, p = 0.0147. N=9 for Control mCherry; N=12 for Rostral medNAcSh ChrimsonR. c. Pie charts showing % of mice showing food intake inhibition (mean Δlick counts non-opto/opto>0) in each group: 42% of ChrimsonR rostral medNAcSh mice, 20% of controls. Data is mean ± SEM. *p<0.05; **p<0.01; ***p<0.001.

(3) Molecular profiling:(a) Stard5 expression is present in both D1- and D2-SPNs; comparisons to bulk calcium signals and quantification of percentages across rostral and caudal cells would be helpful. The authors should establish whether these cells also express SerpinB2, an established marker of LH projecting neurons.

We thank the Reviewer for this relevant point. In the photometry experiments (Figure 7) using Stard5-Flp mice, we acknowledge that the recorded signals reflect a mixed population of D1- and D2-SPNs. Based on quantification in a separate set of brains, we estimate that Stard5 is expressed in a variety of cell types, of which 35% are D1-SPNs and 30% are D2-SPNs (Supplementary Figure S3). While Liu et al. 2024 reported no overlap between Stard5 and Drd2, canonical marker for D2-SPNs, available transcriptomic data (Chen et al. 2021) and our own histological and RNA-based analyses (Figure 6 and Supplementary Figure S3) found Stard5 to be expressed in both D1-SPNs and D2-SPNs. Hence, indeed, Stard5 is a mixed population.

We provide here the quantification of percentages of Stard5 expression across rostral and caudal cells: for instance, in the dorsal rostral medNAcSh, 79% of D1-SPNs and 76% of D2-SPNs express Stard5; in the ventral rostral medNAcSh the percentages are 47% and 55%, whereas the same percentages drop to 39 and 31% in the dorsal caudal medNAcSh and 15% and 20% in the ventral caudal medNAcSh.

As suggested by the Reviewer, we also performed further analysis of the publicly available scRNA-seq dataset from Chen et al. 2021, which shows that 4.4% of all Stard5-expressing cells are also Serpinb2+, while 1.8% of all sequenced NAc cells are Stard5+/Drd1+/Serpinb2+ and 0.21% are Stard5+/Drd2+/Serpinb2+.

(b) Verification of the Stard5-2A-Flp line (specificity, overlap with immunomarkers) should be documented more thoroughly.

We agree with the Reviewer that a more detailed characterization of the Stard5-2A-Flp mouse line would be relevant for the validation of the line.

In our study, we identified Stard5 as a marker gene that enables selective targeting of the rostral medNAcSh, as it is strongly enriched in the rostral medNAcSh (Figure 5-7). Stard5-Flp mice injected with Flp-dependent AAV in rostral medNAcSh, NAc core and dorsal striatum show specific AAV expression only in the rostral medNAcSh (Figure 7).

Moreover, we show that the line is specific as injection of a Flp-dependent AAV in a Stard5-Flp negative line does not lead to expression (Figure 7c).

However, re-analysis of the published scRNA-seq dataset (Chen et al. 2021) indicates that Stard5^+^ cells comprise a heterogeneous population, including D1-SPNs (~35%), D2-SPNs (~30%), local interneurons (~18%), glial cells (~12%), and other cell types (Suppl. Fig. S3).

Together, these data validate the Stard5-2A-Flp line as a spatially specific genetic entry point for the rostral medNAcSh, while highlighting the cellular heterogeneity of Stard5-expressing cells. Given the limited brain material left, we were not able to add additional colocalization analyses with immunomarkers, but agree this would be important to include in future studies.

(c) The molecular analysis is restricted to a small set of genes; broader spatial transcriptomics could uncover additional candidate markers. See also above.

We thank the Reviewer for this suggestion. Broader spatial transcriptomic analyses would indeed be highly valuable for identifying additional candidate markers. Our aim for the present study was to identify molecular landmarks to selectively target the rostral medNAcSh, but in a future study, we would be highly interested in building on our initial findings and providing an exhaustive molecular characterization of the region using spatial transcriptomics. We would be particularly motivated to do so, given the important functional specificity of the rostral NAcSh identified in the present publication.

**Reviewer #2 (Public review):**
Summary:Marinescu et al. combine in vivo imaging with circuit-specific optogenetic manipulation to characterize the anatomic heterogeneity of the medial nucleus accumbens shell in the control of food intake. They demonstrate that the inhibitory influence of dopamine D1 receptor-expressing neurons of the medial shell on food intake decreases along a rostro-caudal gradient, while both rostral and caudal subpopulations similarly control aversion. They then identify Stard5 and Peg10 as molecular markers of the rostral and caudal subregions, respectively. Through the development of a new mouse line expressing the flippase under the promoter of Stard5, they demonstrate that Stard5-positive neurons recapitulate the activity of D1positive neurons of the rostral shell in response to food consumption and aversive stimuli.

We thank Reviewer 2 for the positive feedback, summary of our findings and for the thorough reading and constructive comments on the manuscript, which allowed us to improve the quality of the revised version.

Strengths:This study brings important findings for the anatomical and functional characterization of the brain reward system and its implications in physiological and pathological feeding behavior. It is a well-designed study, technically sound, with clear and reliable effects. The generation of the new Stard5-Flp line will be a valuable tool for further investigations. The paper is very well written, the discussion is very interesting, addresses limitations of the findings, and proposes relevant future directions

We thank Reviewer #2 for their supportive feedback.

Weaknesses:At this stage, identification and characterization of the activity of Stard5-positive neurons is a bit disconnected from the rest of the paper, as this population encompasses both D1- and D2-positive neurons as well as interneurons. While they display a similar response pattern as D1-neurons, it remains to be determined whether their manipulation would result in comparable behavioral outcomes.

We agree that this represents an important limitation of the current study. In our search for molecular markers of the rostral feeding hotspot, we identified Stard5 as a marker enriched in the rostral medNAcSh; however, Stard5 labels a heterogeneous population that includes D1- and D2-SPNs as well as other cell types. While Stard5^+^ neurons display activity patterns similar to D1-SPNs, we acknowledge that whether their direct manipulation would produce comparable behavioral effects to D1-SPNs remains to be determined. Moreover, it remains to be determined how the activity and function of Stard5^+^ neurons compares to D2-SPNs.

To specifically isolate Stard5^+^ D1-SPNs, we generated a Stard5-Flp;Drd1-Cre mouse line via breeding. However, the 4 CreON/FlpON AAVs which we tested exhibited leaky expression, including ectopic expression in Cre-positive but Flp-negative cells. This prevented reliable, cell-type-specific manipulation. We are actively working to overcome this common technical limitation of Flp/Cre AAVs, and these experiments will be addressed in a future study.

**Recommendations for the authors:**

**Editor's note:**
Readers would also benefit from coding individual data points by sex and noting N/sex in the figure legends.

We thank the editor for the note, we have noted in each figure legend the N and sex of the mice.

**Reviewer #1 (Recommendations for the authors):**
(1) Integration of results: The manuscript reads as two partly disconnected halves (functional gradient vs. molecular profiling). A more precise articulation of how the molecular findings (Stard5, Peg10) directly relate to the functional data would improve coherence.

We thank the Reviewer for raising this important point. We agree that clearer integration between the functional gradient and the molecular findings would strengthen the manuscript. In the present study, Stard5 and Peg10 are not introduced as mechanistic drivers of behavior, but as molecular landmarks that map onto the functional rostro-caudal organization of the medNAcSh.

Stard5 expression is enriched in the rostral medNAcSh, where we identify a functional hotspot for rewardrelated feeding, whereas Peg10 marks more caudal territories. Thus, the molecular profiling provides an independent axis that aligns with and supports the functional gradient revealed by photometry and optogenetic experiments. Whether these genes themselves contribute causally to feeding or aversive behaviors remains an open and interesting question for future studies.

To improve clarity, we have explicitly articulated this link in the Discussion:

“Importantly, our results indicate that spatial organization also defines functional specialization in the medNAcSh, and that molecular markers such as Stard5 provide access to these spatially defined subterritories rather than labeling a single, homogenous neuronal subtype.“

“Having established a robust functional dichotomy of D1-SPNs along the rostro-caudal axis in reward consumption, we next asked whether this functional organization is mirrored by differences in molecular composition across the medNAcSh. Using multiple anatomical techniques, we find strong differences in the molecular composition of the rostral vs. caudal medNAcSh, which in turn could explain behavioral differences between these brain subregions.”

“This makes Stard5 a spatial molecular landmark that captures the cellular ensemble of the rostral feeding hotspot, rather than a marker defining a single functional cell class. It is interesting that Stard5, a STARTdomain protein implicated in cholesterol metabolism and cellular stress responses (Alpy and Tomasetto, 2005; Rodriguez-Agudo et al., 2012; Calderon-Dominguez et al., 2014), and Peg10, an imprinted gene with roles in embryonic development and cancer (Mou et al. 2025), mark distinct rostro-caudal domains of the medNAcSh. Whether these genes themselves causally contribute to appetitive and consummatory behaviors, or aversive processing in this region remains an important question for future studies.”

(2) Injection site specificity: Given prior work on NAc manipulations, it is essential to ensure precise targeting. Representative images from both rostral and caudal placements, including verification of fiber/injection confinement, would increase confidence.

We thank the Reviewer for this important point regarding injection site specificity. Optic fiber placement was validated by identifying the coronal section in which the fiber tip was centered and aligning it to the mouse brain atlas (Franklin and Paxinos, The Mouse Brain in Stereotaxic Coordinates). We validated currently a total of 14 brains, shown in the newly added Supplementary Figure S10.

The primary source of variability across animals could be the extent of the viral spread and the size of the optic implants, which were 400 for photometry experiments and 200 μm for the optogenetic studies. We acknowledge that this limits the spatial precision with which the individual subregions can be isolated. This limitation is explicitly discussed in the manuscript.

Importantly, despite this limitation, we detected robust and reproducible differences between rostral and caudal medNAcSh in reward-consumption photometry and optogenetic assays. This argues against injection site proximity or fiber misplacement being a major confounding factor for the main conclusions. Nonetheless this comment is a valid point, and in future studies we plan to establish targeting methods with reduced viral volumes and/or tapered optic fibers (Pisanello et al. 2017). This will allow finer spatial restriction and more precise dissection of medNAcSh subregions.

(3) Minor clarifications:(a) Provide explicit definitions of "rostral" and "caudal" coordinates.

We adjusted Figure 1 and added the coordinates.

(b) Consider alternative wording to "gradient" since only two rostro-caudal positions are tested.

RNA-seq and MERFISH data indicate that molecular markers in the NAcSh are organized along a continuous rostro–caudal gradient rather than discrete boundaries (Chen et al. 2021; Stanley et al. 2020). Our use of the term ‘gradient’ therefore reflects this established molecular organization, even though our functional experiments sampled two representative positions along this continuum.

We added the following sentence in the discussion for clarification:

“Of note, in this paper we decided to use the term “rostro-caudal gradient”, motivated by converging evidence from prior pharmacological studies (see below) and scRNA sequencing data (Chen et al., 2021; Stanley et al., 2020), which show continuous molecular and functional changes along the rostro-caudal axis of the medNAcSh rather than sharply defined boundaries. Our use of the term ‘gradient’ therefore reflects this established molecular organization, even though our functional experiments sampled only two representative positions along this continuum.”

(c) Enhance representative images (e.g., stronger DAPI, zoom-ins, bregma coordinates).

To improve clarity, we have adjusted Figure 1 by adding schematic representations including stereotaxic surgery coordinates, which facilitate interpretation of rostro–caudal targeting.

(d) Report trial numbers in figure legends, injection site details (e.g., S1 mouse), learning curves, and rationale for low-pass filtering in photometry.

We thank the Reviewer for these suggestions. The average number of successful trials is now reported in the figure legends (Figure 1 and Figure 7). Injection site details are described in the Methods and are now also illustrated in Figure 1a and validated in Supplementary Figure S10. In addition, we have added Supplementary Figure S8 showing the learning curves of the Drd1-Cre and Stard5-Flp mice included in this study.

Regarding the low-pass filtering in photometry analysis: low-pass filtering (1 Hz) was applied to the signal to remove high-frequency noise and isolate slow calcium-dependent fluorescence fluctuations that reflect population-level neural activity as we have done before (Labouesse et al. 2023, 2024). Low-pass filtering is a commonly-used analysis in fiber photometry and often shows a better artifact-corrected signal (Zhang et al. 2023; Keevers and Jean-Richard-dit-Bressel 2025).

**Reviewer #2 (Recommendations for the authors):**
Major Comments:(1) As mentioned, I find the part on Stard5-positive neurons a bit disconnected. Ideally, as mentioned in the discussion, the author could cross Stard5-Flp mice with D1-cre to selectively monitor and/or manipulate these neurons. Alternatively, do they have any data regarding D2-positive neurons of the rostral part to show whether they behave differently from D1-positive neurons?

We thank the Reviewer for this suggestion and agree that selectively monitoring or manipulating Stard5^+^ D1-SPNs using an intersectional approach would strengthen the link between the molecular and functional findings. We are pursuing this strategy by crossing Stard5-Flp mice with Drd1-Cre mice; however, as noted above, currently available CreON/FlpON viral tools exhibited leaky expression (a commonly known problem for such AAVs), preventing reliable cell-type–specific targeting. As a result, these experiments are ongoing (including reducing the titers) and will be addressed in a future study.

At present, we do not have equivalent functional data for D2-SPNs in the rostral medNAcSh. Investigating whether rostral D2-SPNs behave differently from caudal D2-SPNs is an important and interesting question, which we hope to address in a future study. This limitation is acknowledged in the discussion.

(2) Do the authors have any data on locomotor activity when they manipulate D1-expressing neurons? Lower food consumption as well as lower activity in the stimulated compartment - interpreted as aversion - could be related to diminished locomotor activity.

We thank the reviewer for the relevant point about locomotion. We ran new analyses of locomotor activity during the feeding task (operant boxes) using a machine-learning model. A small subset of frames (136 frames from 10 video recordings) was manually annotated to define the animal’s body center and nose, as well as the four corners of the operant box. These annotations were used to train a YOLO (Redmon et al. 2015)-based pose estimation model. Locomotion metrics, such as total distance moved were subsequently derived from the temporal integration of positional data and aligned to opto-on and opto-off epochs of the feeding task. During licking periods, the animal’s body center remains largely stationary, which could lead to an overestimation of immobility. Nevertheless, we quantified the total distance traveled in the entire operant box across epochs, shown in Supplementary Figure S9 a-b. In our proof-of-concept experiment (Figure 2c-e), locomotion was increased in rostral ChrimsonR mice compared to controls (Supplementary Figure S9a), a similar effect seen with chemogenetic activation of D1-SPNs (Zhu, Ottenheimer, and DiLeone 2016). In our full experimental cohort, locomotion did not differ between control, rostral and caudal ChrimsonR mice across laser on and laser off epochs. These results indicate that reduced reward consumption during stimulation of rostral D1-SPNs is not due to decreased locomotor activity. Notably, whereas the inhibitory effect on consumption is specific to rostral D1-SPNs activation, locomotor effects are similar for both rostral and caudal D1-SPNs stimulation, indicating they are at least partly dissociated from one another.

Moreover, in the RTPPA task, it is accepted that the percentage of time spent in the light-paired chamber reflects the preference or aversiveness to optogenetic stimulation. We additionally quantified total distance traveled (Supplementary Figure S9c). While optogenetic stimulation of both rostral and caudal D1-SPNs reduced time spent in the light-paired chamber (Figure 4), total distance traveled was unchanged, indicating that the observed aversion is not due to reduced locomotion.

We added the following to the Results section: “To determine whether the reduced reward consumption observed in Rostral ChrimsonR mice could be explained by changes in locomotion, we quantified the total distance traveled during this task. Optogenetic stimulation led to an increase in locomotion in the small cohort of Rostral ChrimsonR mice in the reward consumption experiment shown in Figure 2d-e (Supplementary Figure S9a), while no change in locomotion was observed across epochs in mCherry controls, ChrimsonR Rostral and Caudal mice (Supplementary Figure S9b, related to Figure 2g-i)”

And

“Quantification of locomotion showed no reduction in distance traveled in the light-paired chamber (Supplementary Figure S9c), indicating that the avoidance was not driven by impaired locomotion. These data indicate that medNAcSh D1-SPNs generally promote aversion without affecting locomotion and without major differences along the rostro-caudal axis”

Additionally, we added the following sentence to the Discussion: “Importantly, our behavioral effects of rostral D1-SPNs in the reward consumption and RTTPA assays could not be explained by reduced locomotor activity. Indeed, optogenetic stimulation of D1-SPNs during the reward consumption task did not reduce locomotion; instead, locomotion was either unchanged or increased in a small cohort of Rostral ChrimsonR mice. The increased locomotion likely reflected appetitive behavior and is consistent with past chemogenetic studies (Zhu et al., 2016). In the RTTPA no locomotion differences were detected.“

(3) It would be useful to provide a schematic (or pictures) for the location of fiber implantation in all animals for both photometry and optogenetics.

We validated optic fiber placement in 14 animals by identifying the coronal section in which the fiber tip was centered and aligning this section to the mouse brain atlas (Franklin and Paxinos, The Mouse Brain in Stereotaxic Coordinates). Representative optic fiber placement and viral spread are shown in the newly added Supplementary Figure S10.

Minor Comments:(1) Figure 6e and g seem mislabeled: "Drd1+ (D2-SPNs)".

Yes, thank you. We corrected it.

(2) Line 395-397: the authors mention Flp minimal Flp Leakage, but could it be low activity of Stard5 promoter in the core and dorsal striatum that allows little expression of the flippase that could be sufficient for recombination?

We thank the Reviewer for this insightful point. We cannot fully distinguish between these possibilities in the current study; however, the overall recombination outside the target region remains minimal, supporting the utility of the Stard5-Flp line for selective targeting of the rostral medNAcSh. Injection of a Flp-dependent AAV into the lateral shell, core and dorsal striatum showed no expression, therefore we think this is unlikely. Moreover, this aligns with Stard5 expression patterns derived from the scRNAseq data (Chen et al. 2021), Allen Brain Atlas quantifications (Figure 5) and our RNAscope analysis (Figure 6). Nevertheless, we acknowledge that histology alone cannot definitively exclude this possibility, and quantitative approaches such as qPCR would be required.

References

Alpy, Fabien, and Catherine Tomasetto. 2005. “Give Lipids a START: The StAR-Related Lipid Transfer (START) Domain in Mammals.” Journal of Cell Science 118(13):2791–2801. doi:10.1242/jcs.02485.

Calderon-Dominguez, Maria, Gregorio Gil, Miguel Angel Medina, William M. Pandak, and Daniel RodríguezAgudo. 2014. “The StarD4 Subfamily of Steroidogenic Acute Regulatory-Related Lipid Transfer (START) Domain Proteins: New Players in Cholesterol Metabolism.” The International Journal of Biochemistry & Cell Biology 49:64–68. doi:10.1016/j.biocel.2014.01.002.

Chen, Renchao, Timothy R. Blosser, Mohamed N. Djekidel, Junjie Hao, Aritra Bhattacherjee, Wenqiang Chen, Luis M. Tuesta, Xiaowei Zhuang, and Yi Zhang. 2021. “Decoding Molecular and Cellular Heterogeneity of Mouse Nucleus Accumbens.” Nature Neuroscience 24(12):1757–71. doi:10.1038/s41593-021-00938-x.

Domingues, Ana Verónica, Tawan T. A. Carvalho, Gabriela J. Martins, Raquel Correia, Bárbara Coimbra, Ricardo Bastos-Gonçalves, Marcelina Wezik, Rita Gaspar, Luísa Pinto, Nuno Sousa, Rui M. Costa, Carina Soares-Cunha, and Ana João Rodrigues. 2025. “Dynamic Representation of Appetitive and Aversive Stimuli in Nucleus Accumbens Shell D1- and D2-Medium Spiny Neurons.” Nature Communications 16(1):59. doi:10.1038/s41467-024-55269-9.

Keevers, Luke J., and Philip Jean-Richard-dit-Bressel. 2025. “Obtaining Artifact-Corrected Signals in Fiber Photometry via Isosbestic Signals, Robust Regression, and DF/F Calculations.” Neurophotonics 12(02). doi:10.1117/1.NPh.12.2.025003.

Labouesse, Marie A., Arturo Torres-Herraez, Muhammad O. Chohan, Joseph M. Villarin, Julia Greenwald, Xiaoxiao Sun, Mysarah Zahran, Alice Tang, Sherry Lam, Jeremy Veenstra-VanderWeele, Clay O. Lacefield, Jordi Bonaventura, Michael Michaelides, C. Savio Chan, Ofer Yizhar, and Christoph Kellendonk. 2023. “A Non-Canonical Striatopallidal Go Pathway That Supports Motor Control.” Nature Communications 14(1):6712. doi:10.1038/s41467-023-42288-1.

Labouesse, Marie A., Maria Wilhelm, Zacharoula Kagiampaki, Andrew G. Yee, Raphaelle Denis, Masaya Harada, Andrea Gresch, Alina-Măriuca Marinescu, Kanako Otomo, Sebastiano Curreli, Laia Serratosa Capdevila, Xuehan Zhou, Reto B. Cola, Luca Ravotto, Chaim Glück, Stanislav Cherepanov, Bruno Weber, Xin Zhou, Jason Katner, Kjell A. Svensson, Tommaso Fellin, Louis-Eric Trudeau, Christopher P. Ford, Yaroslav Sych, and Tommaso Patriarchi. 2024. “A Chemogenetic Approach for Dopamine Imaging with Tunable Sensitivity.” Nature Communications 15(1):5551. doi:10.1038/s41467-024-49442-3.

Liu, Yiqiong, Ying Wang, Zheng-dong Zhao, Guoguang Xie, Chao Zhang, Renchao Chen, and Yi Zhang. 2024. “A Subset of Dopamine Receptor-Expressing Neurons in the Nucleus Accumbens Controls Feeding and Energy Homeostasis.” Nature Metabolism 6(8):1616–31. doi:10.1038/s42255-02401100-0.

Mou, Dachao, Shasha Wu, Yanqiong Chen, Yun Wang, Yufang Dai, Min Tang, Xiu Teng, Shijun Bai, and Xiufeng Bai. 2025. “Roles of PEG10 in Cancer and Neurodegenerative Disorder (Review).” Oncology Reports 53(5):1–9. doi:10.3892/or.2025.8893.

O’Connor, Eoin C., Yves Kremer, Sandrine Lefort, Masaya Harada, Vincent Pascoli, Clément Rohner, and Christian Lüscher. 2015. “Accumbal D1R Neurons Projecting to Lateral Hypothalamus Authorize Feeding.” Neuron 88(3):553–64. doi:10.1016/j.neuron.2015.09.038.

Pedersen, Christian E., Raajaram Gowrishankar, Sean C. Piantadosi, Daniel C. Castro, Madelyn M. Gray, Zhe C. Zhou, Shane A. Kan, Patrick J. Murphy, Patrick R. O’Neill, and Michael R. Bruchas. 2022. “Medial Accumbens Shell Spiny Projection Neurons Encode Relative Reward Preference.”

Pisanello, Ferruccio, Gil Mandelbaum, Marco Pisanello, Ian A. Oldenburg, Leonardo Sileo, Jeffrey E. Markowitz, Ralph E. Peterson, Andrea Della Patria, Trevor M. Haynes, Mohamed S. Emara, Barbara Spagnolo, Sandeep Robert Datta, Massimo De Vittorio, and Bernardo L. Sabatini. 2017. “Dynamic Illumination of Spatially Restricted or Large Brain Volumes via a Single Tapered Optical Fiber.” Nature Neuroscience 20(8):1180–88. doi:10.1038/nn.4591.

Redmon, Joseph, Santosh Divvala, Ross Girshick, and Ali Farhadi. 2015. “You Only Look Once: Unified, Real-Time Object Detection.”

Requejo-Mendoza, Nikte, José-Antonio Arias-Montaño, and Ranier Gutierrez. 2025. “Nucleus Accumbens D2-Expressing Neurons: Balancing Reward and Licking Disruption through Rhythmic Optogenetic Stimulation” edited by J. M. Dominguez. PLOS ONE 20(2):e0317605. doi:10.1371/journal.pone.0317605.

Rodriguez-Agudo, Daniel, Maria Calderon-Dominguez, Miguel Angel Medina, Shunlin Ren, Gregorio Gil, and William M. Pandak. 2012. “ER Stress Increases StarD5 Expression by Stabilizing Its MRNA and Leads to Relocalization of Its Protein from the Nucleus to the Membranes.” Journal of Lipid Research 53(12):2708–15. doi:10.1194/jlr.M031997.

Stanley, Geoffrey, Ozgun Gokce, Robert C. Malenka, Thomas C. Südhof, and Stephen R. Quake. 2020. “Continuous and Discrete Neuron Types of the Adult Murine Striatum.” Neuron 105(4):688-699.e8. doi:10.1016/j.neuron.2019.11.004.

Zhang, Yan, Márton Rózsa, Yajie Liang, Daniel Bushey, Ziqiang Wei, Jihong Zheng, Daniel Reep, Gerard Joey Broussard, Arthur Tsang, Getahun Tsegaye, Sujatha Narayan, Christopher J. Obara, JingXuan Lim, Ronak Patel, Rongwei Zhang, Misha B. Ahrens, Glenn C. Turner, Samuel S. H. Wang, Wyatt L. Korff, Eric R. Schreiter, Karel Svoboda, Jeremy P. Hasseman, Ilya Kolb, and Loren L. Looger. 2023. “Fast and Sensitive GCaMP Calcium Indicators for Imaging Neural Populations.” Nature 615(7954):884–91. doi:10.1038/s41586-023-05828-9.

Zhu, Xianglong, David Ottenheimer, and Ralph J. DiLeone. 2016. “Activity of D1/2 Receptor Expressing Neurons in the Nucleus Accumbens Regulates Running, Locomotion, and Food Intake.” Frontiers in Behavioral Neuroscience 10. doi:10.3389/fnbeh.2016.00066.